

# On four-point connectivities in the critical 2d Potts model

Marco Picco[1*], Sylvain Ribault[2] and Raoul Santachiara[3]

**1** LPTHE, Sorbonne Université and CNRS, France
**2** Institut de physique théorique, CNRS, CEA, Université Paris-Saclay, France
**3** LPTMS, CNRS (UMR 8626), Université Paris-Saclay, 91405 Orsay, France

$\star$ picco@lpthe.jussieu.fr

## Abstract

We perform Monte-Carlo computations of four-point cluster connectivities in the critical 2d Potts model, for numbers of states $Q \in (0, 4)$ that are not necessarily integer. We compare these connectivities to four-point functions in a CFT that interpolates between D-series minimal models. We find that 3 combinations of the 4 independent connectivities agree with CFT four-point functions, down to the 2 to 4 significant digits of our Monte-Carlo computations. However, we argue that the agreement is exact only in the special cases $Q = 0, 3, 4$. We conjecture that the Potts model can be analytically continued to a double cover of the half-plane $\{\Re c < 13\}$, where $c$ is the central charge of the Virasoro symmetry algebra.



# 1 Introduction and summary

## 1.1 Why this model, why these connectivities?

**Why the critical Potts model**

The Potts model is a statistical model that is simple to define, yet rich enough for describing interesting phenomenons such as percolation, in particular thanks to the presence of a tunable parameter $Q$. The $Q$-state Potts model was originally defined in terms of spins that could take $Q$ values, where $Q \geq 2$ is a natural integer. Fortuyn and Kasteleyn's reformulation in terms of clusters [1] allows $Q$ to take more generic values, possibly $Q \in \mathbb{C}$. We will see that studying the $Q$-dependence of observables can be fruitful, and that some questions are easier to address at non-integer values of $Q$.

We focus on the limit of infinite lattice size, and further assume that the bond probability takes a particular critical value, such that the model becomes conformally invariant. This allows us to use the conformal bootstrap method, and to obtain theoretical predictions that can be compared to numerical simulations of the model. Therefore, the critical limit gives insight into the qualitative behaviour of the model, and helps validate our numerical approach.

**Why four-point connectivities**

In order to build observables that are defined for non-integer $Q$, we should focus on clusters rather than on spins. We will therefore consider cluster connectivities: probabilities that a number of points belong to the same cluster, or to different clusters. Although connectivities

are non-local quantitites, we work under the basic assumption that cluster connectivities are linearly related to correlation functions of local fields in a CFT that describes the critical Potts model. This assumption is known to hold when the clusters are anchored to a boundary [2–5] and in some other cases [6,7]. In particular, the three-point connectivity of the two-dimensional Potts model was found to be related to a three-point function in Liouville theory [6].

In order to understand the CFT, we need to compute four-point correlation functions. Unlike two- and three-point functions, a four-point function has a nontrivial dependence on the four points' positions. Moreover, a four-point function encodes the model's spectrum, in the sense that the $s$-, $t$- and $u$-channel decompositions of a four-point function are sums over subsets of the spectrum. Depending on the context, we will use the word spectrum for the full spectrum of a model, or for the spectrum of a four-point function in a particular channel.

The numbers $F_n$ of independent $n$-point connectivities, i.e. the numbers of partitions of $n$ into subsets of size at least two, are [7]

$$F_2 = 1 \ , \ F_3 = 1 \ , \ F_4 = 4 \ , \ F_5 = 11 \ , \ F_6 = 41 \ , \ F_7 = 162 \ , \dots \ . \tag{1}$$

These numbers have no known interpretation in terms of the combinatorics of correlation functions. However, for a given number of points such as $n = 4$, we can look for correlation functions that have the same behaviour as connectivities under permutations and under conformal transformations. This already provides strong contraints on possible relations between connectivities and correlation functions.

### Why in two dimensions

Two-dimensional CFT is easier than higher-dimensional CFT, because there are infinitely many independent conformal transformations in $d = 2$, versus finitely many for $d > 2$. Some nontrivial two-dimensional CFTs are exactly solvable, for example minimal models or Liouville theory. In the Potts model itself, the dimensions of some operators are exactly known in $d = 2$ only, although such exact results fall far short of a complete solution of the model. We view our work as a step towards exactly solving the two-dimensional Potts model.

In two dimensions, conformal symmetry is described by the Virasoro algebra, which has a paramter $c$ called the central charge. The central charge is related to the number $Q$ of states of the Potts model via [8]

$$Q = 4\cos^2 \pi\beta^2 \quad \text{with} \quad \tfrac{1}{2} \leq \Re\beta^2 \leq 1 \ , \qquad c = 1 - 6\left(\beta - \beta^{-1}\right)^2 \ . \tag{2}$$

## 1.2 State of the art and open problems

### Monte-Carlo computations and the bootstrap approach

In our previous work [9], we have performed Monte-Carlo computations of four-point connectivities, with sufficient precision for being tested against CFT predictions or guesses. Moreover, we have introduced a numerical implementation of the conformal bootstrap approach that tests whether an exact ansatz for the spectrum gives rise to crossing-symmetric four-point functions. In principle, this reduces the computation of four-point functions to the determination of the exact spectrum.

### The partition function and the spectrum

What do we know of the spectrum of the Potts model? Although the torus partition function was computed in the classic article [8] (reviewed in [10]), this is insufficient for determining the spectrum, for at least two reasons:

1. The partition function is a combination of characters of representations, but two different representations can have the same character. (Representations where the Virasoro generator $L_0$ is not diagonalizable are not fully characterized by their $L_0$ eigenvalues and therefore by their characters.)

2. The decomposition of the partition function into characters involves multiplicities that need not be positive integers. A character might therefore be absent because several contributions of the same representation cancel. And we cannot distinguish the contribution $\chi_{R/R'}$ of a quotient representation, from the contribution $\chi_R - \chi_{R'}$ of two representations.

In spite of these caveats, the partition function provides useful hints on which representations may appear in the spectrum, and we will denote $\mathcal{S}^{\text{Potts}}$ the smallest set of representations that is compatible with the partition function at generic $Q$.

**A simple ansatz for the spectrum**

In [9], we have proposed a simple ansatz $\mathcal{S}_{2\mathbb{Z},\mathbb{Z}+\frac{1}{2}}$ for the spectrum. This ansatz leads to crossing-symmetric four-point functions, so it must be the spectrum of a consistent CFT, which was later called the **odd CFT** in [11]. This CFT may seem to describe the Potts model, based on the numerical agreement of four-point functions with certain specific linear combinations of connectivities. However, $\mathcal{S}_{2\mathbb{Z},\mathbb{Z}+\frac{1}{2}}$ is actually a small subset of $\mathcal{S}^{\text{Potts}}$, raising doubts that our four-point functions exactly coincide with connectivities. In an attempt at settling the issue [12], Jacobsen and Saleur have used an independent method based on lattice algebras, and found numerical evidence that connectivities involve states from $\mathcal{S}^{\text{Potts}}$ that are absent from $\mathcal{S}_{2\mathbb{Z},\mathbb{Z}+\frac{1}{2}}$. However, this statement is only about connectivities, and does not say how they are related to four-point functions in a consistent CFT.

Both our Monte-Carlo calculations, and the lattice algebraic methods of [12], are currently sensitive to only a few low-lying states in the spectrum. This is barely enough for seeing a difference between $\mathcal{S}_{2\mathbb{Z},\mathbb{Z}+\frac{1}{2}}$ and $\mathcal{S}^{\text{Potts}}$, and insufficient for probing $\mathcal{S}^{\text{Potts}}$ in detail. A much stronger test could come from probing crossing symmetry of four-point functions built from $\mathcal{S}^{\text{Potts}}$, using the techniques of [9]. But this would first require determining the full structure of the representations, not just their characters.

In the present work we will further investigate the agreement between Monte-Carlo computations, and the odd CFT. We will also discuss the agreement in light of recent progress in the analytic understanding of the odd CFT. We will show analytically that our correlation functions must differ from connectivities, and explain why the difference is so small.

## 1.3 Results

**Progress in understanding the CFT**

The spectrum $\mathcal{S}_{2\mathbb{Z},\mathbb{Z}+\frac{1}{2}}$ was originally introduced as a guess that led to crossing-symmetric four-point functions, based on a numerical bootstrap analysis [9]. We now know that $\mathcal{S}_{2\mathbb{Z},\mathbb{Z}+\frac{1}{2}}$ is the non-diagonal sector of the odd CFT: an exactly solvable CFT that can be viewed as a limit of D-series minimal models [13].

By exactly solvable, we mean a CFT whose structure constants are known analytically, allowing us to compute four-point functions with an arbitrary precision. Having a high precision on the CFT side makes no difference in the comparison with cluster connectivities, since numerical bootstrap results were already more precise than Monte-Carlo computations. The interest of having an exactly solvable CFT is rather that we can study its features qualitatively.

The behaviour of the odd CFT when the central charge becomes rational was investigated in [11]. Two striking features were found:

1. For certain rational values of the central charge, the odd CFT reduces to a D-series minimal model. When this minimal model is not unitary, its spectrum contains states with negative conformal dimensions. Such states contribute to our four-point functions in the channel where the spectrum is not $\mathcal{S}_{2\mathbb{Z},\mathbb{Z}+\frac{1}{2}}$.

2. As functions of the central charge, our four-point functions have poles at certain rational values, and these rational values are dense in the interval $c \in (-2, 1)$ that corresponds to $Q \in (0, 4)$.

The negative dimensions contradict the expected large distance decay of cluster connectivities, and show that the agreement with our four-point functions cannot be exact. However, these dimensions are $\geq -\frac{1}{24}$ for $Q \in (0, 4)$, and therefore hard to distinguish from 0 numerically. Similarly, the poles of four-point functions are hard to see numerically: most of them are eliminated by the numerical truncations that we use, and the few surviving poles have small residues. The only pole that is too strong to ignore is the pole at $Q = 2$, which can however be absorbed in a coefficient of the linear relation between cluster connectivities and four-point functions. It is very plausible that cluster connectivities are smooth functions of $Q$ and cannot have poles, but it is surprisingly hard to numerically see any difference between smooth connectivities, and singular correlation functions.

On the other hand, the spectrum $\mathcal{S}^{\text{Potts}}$ does not have states with negative dimensions, and it contains states whose contributions can potentially cancel all the poles of four-point functions, as was sketched in an example in [12]. So it is a better candidate for the spectrum of the Potts model. But it remains to show that the corresponding four-point functions are crossing-symmetric, to compute them with a reasonable precision, and if possible to exactly solve the model. All this will be much harder than with the spectrum $\mathcal{S}_{2\mathbb{Z},\mathbb{Z}+\frac{1}{2}}$.

**Monte-Carlo computations: challenges and results**

The critical Potts model in principle lives on an infinite, smooth plane, while our Monte-Carlo computations are done on a finite periodic lattice, and depend on a scale $r$ and lattice size $L$. Our first task is therefore to find the scaling limit $1 \ll r \ll L$ of our data, by fitting them to appropriate ansatzes for their dependence on $r$ and $L$. Our ansatzes (58) are inspired by CFT on a torus, and by the analysis of the numerical data: in particular, we find that the correction to the scaling limit that matters most is a topological correction induced by the energy operator.

Once we have robust estimates for the scaling limits of connectivities, we evaluate the numerical error by comparing with exactly known results for $Q = 2, 3, 4$. We find that errors increase strongly with $Q$: we have 4 significant digits at $Q = 2$, only 3 significant digits at $Q = 3$, and poor accuracy at $Q = 4$. We therefore know when deviations from theoretical predictions should be attributed to numerical errors.

We then compare linear combinations of the four basic four-point connectivities $P_0, P_1, P_2, P_3$, with three four-point functions in the odd CFT $R_1, R_2, R_3$. We find the relation (61), whose coefficients are simple analytic functions of $Q$. The relation even works for values of $Q$ where $R_\sigma$ has a pole while $P_\sigma$ is smooth, because the poles are truncated out when we numerically evaluate $R_\sigma$ using its decomposition into conformal blocks. Of course, a more precise determination of $P_\sigma$ would force us to keep more terms in $R_\sigma$, and some poles would survive, leading to violations of the relation near these poles.

After comparing our connectivities with the odd CFT, we use them for extracting the first 3 to 4 lowest-lying states of the Potts model's spectrum, and the corresponding structure constants. This involves fitting the Monte-Carlo data to truncated conformal block expansions. We compare the results with predictions from two main sources:

1. General properties of connectivities, as reviewed in Section 3.1,

2. the spectrum $\mathcal{S}^{\text{Potts}}$, as summarized in Eq. (27).

As we summarize in Eqs. (68) and (3.6), the results uniformly agree with the predictions. We emphasize that we are comparing functions of $Q$ and not just numbers; structure constants and not just their ratios. (See Appendix A for how structure constants are normalized.)

**Focus on $Q = 0, 3, 4$**

For $Q = 0, 3, 4$, the Potts model simplifies considerably, and some connectivities can be exactly computed. We will show that in these three cases, the relation (61) between four-point functions and connectivities becomes exact.

For $Q = 0$, the Potts model reduces to a free fermionic theory, and the odd CFT reduces to a kind of minimal model with an empty Kac table. For $Q = 3$, the three relevant combinations of connectivities reduce to correlation functions of spin operators, which are given by a D-series minimal model – and the odd CFT reduces to that same D-series minimal model. This minimal model has an alternative description in terms of the $\mathcal{W}_3$ symmetry algebra, whose structure incorporates the $\mathbb{Z}_3$ symmetry between the 3 states of the Potts model. For $Q = 4$, the Potts model reduces to an Ashkin–Teller model, which can be solved thanks to its description as an orbifold of a free CFT.

## 1.4 Conclusion and outlook

**An exactly solvable CFT that approximates four-point connectivities**

We have shown that 3 combinations of the 4 four-point connectivities in the Potts model are approximately related to four-point functions in an exactly solvable CFT. The approximation is good enough that we cannot reach its limits with Monte-Carlo calculations. This relation can be useful, because it comes with quick and easy computations of four-point functions, including for values of $Q$ that are hard to reach otherwise. The relation becomes exact for $Q = 0, 3, 4$ and we give analytic evidence that it is not exact otherwise. This comes in addition to the numerical evidence of [12]. Even when the relation is not exact, the 3 combinations of connectivities that we consider may well have special properties, such as particularly sparse spectrums, and particularly simple structure constants.

This leads to the natural issue of describing the fourth connectivity, i.e. the connectivity that does not appear in our relation. In the odd CFT, the natural ansatz for the fourth connectivity would be a four-point function in Liouville theory, but the comparison with the Monte-Carlo computations fails in this case. The fourth connectivity is only known in the case $Q = 4$, but it does not belong to the odd CFT in that case. Actually, the odd CFT is unable to describe not only the fourth four-point connectivity, but also most $N \geq 5$-point connectivities. This is because the odd CFT contains only two primary fields with the right conformal dimension for describing cluster connectivities, and the number of independent $N$-point functions of these primary fields is less than the number (1) of independent $N$-point connectivities if $N \geq 5$. In the limit $N \to \infty$, we would in fact need infinitely many different primary fields with the right conformal dimension.

We find it remarkable that our four-point functions can come so close to combinations of connectivities, without actually coinciding. If the connectivities are four-point functions in a consistent CFT, this means that the space of consistent CFTs is complicated, and that it may be difficult to chart this space using numerical bootstrap techniques that focus on the contributions of a few low-lying states to a few four-point functions.

**A long road to solving the Potts model**

If there exists a consistent CFT whose spectrum is $\mathcal{S}^{\text{Potts}}$, that CFT will certainly be much harder to solve than the odd CFT. The solution of the odd CFT relies on the existence of two independent degenerate fields, and this is reflected in the structure of the spectrum $\mathcal{S}_{2\mathbb{Z}, \mathbb{Z}+\frac{1}{2}}$, whose states are labelled by two integers [13]. On the other hand, states in $\mathcal{S}^{\text{Potts}}$ are labelled by one integer and one fraction, so we expect that only one degenerate field exists. This is insufficient for completely determining structure constants using known analytic bootstrap techniques. A different source of information is the assumption that four-point functions are smooth functions of $Q$: as sketched in [12], cancellations of poles then provide analytic constraints, which are however far from enough for determining structure constants. Further analytic constraints on structure constants, such as Eq. (70), can also be derived from extra assumptions on the combinations of connectivities that appear in our relation (61).

To summarize, we can see several important open problems on the road to solving the Potts model:

- determining the relation between correlation functions and connectivities,

- determining the spectrum and fusion rules,

- computing four-point connectivities via numerical bootstrap techniques,

- analytically determining the structure constants.

**General complex values of $Q$**

Neither in the Fortuyn–Kasteleyn formulation of the Potts model, nor in the odd CFT, is there any reason for $Q$ to be a real number. The odd CFT actually exists for all central charges such that $\Re c < 13$ i.e. $\Re \beta^2 > 0$ [9, 13], and this more than covers the whole complex $Q$-plane via the map (2). The map is not one-to-one, and in particular each value of $Q \in (4, \infty)$ corresponds to two complex conjugate values of $c$, giving rise to the phenomenon of walking renormalization group flows [10]. See Figures 1.1 and 1.2 for plots of the complex $Q$-plane and of the corresponding values of $\beta^2$ and $c$.

Notice that two different values of $Q$ can lead to the same central charge, due to $c(\beta) = c(\beta^{-1})$. However, the two corresponding CFTs are not the same, because the duality $\beta \to \beta^{-1}$ amounts to changing the spectrum as $\mathcal{S}_{2\mathbb{Z}, \mathbb{Z}+\frac{1}{2}} \to \mathcal{S}_{\mathbb{Z}+\frac{1}{2}, 2\mathbb{Z}}$. This changes the odd CFT into a different model called the even CFT [11]. Starting from a value of $\beta$, the dual value is obtained by analytic continuation around $c = 1$.

One may wonder what happens if we continue the critical Potts model to central charges that do not correspond to any value of $Q$. The answer is known to some extent, as the segment $1 < \beta^2 < \frac{3}{2}$ corresponds to the tricritical Potts model [10]. More generally, the model is critical or multicritical depending on the number of relevant scalar fields [14], in other words the number of diagonal primary states with dimensions such that $\Re \Delta < 1$. The spectrum of the Potts model is believed to include degenerate diagonal primary states with dimensions $\Delta_{(r,1)}$ (5) for $r \in \mathbb{N}^*$ [12], and we have

$$\Re \Delta_{(r,1)} < 1 \iff \Re \beta^2 < \frac{2}{r-1} \ . \tag{3}$$

The region $\frac{1}{2} < \Re \beta^2 < 1$ therefore corresponds to the field $(3, 1)$ being relevant while $(5, 1)$ is irrelevant. The relevance of other fields, starting with $(4, 1)$, seems less important.

We conjecture that the Potts model can be analytically continued to the same half-plane $\{\Re \beta^2 > 0\} = \{\Re c < 13\}$ as the odd CFT, even though its interpretation may change when we leave the region that corresponds to $\frac{1}{2} \le \Re \beta^2 \le 1$ (2). Due to the duality $\beta \to \beta^{-1}$, we expect that the model lives on a double cover of that half-plane.

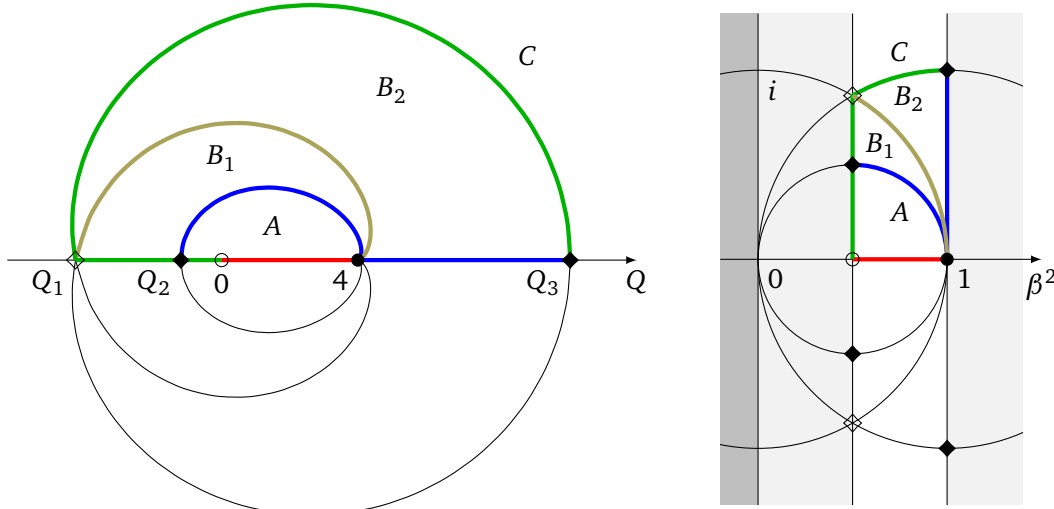

Figure 1.1: The complex $Q$-plane, and the corresponding strip (white) in the complex $\beta^2$-plane according to Eq. (2). The rest of the $\beta^2$-plane is in gray, or dark gray for the region $\{\Re\beta^2 \leq 0\}$ where the odd CFT does not exist. Special values of $Q$ and $\beta^2$ are listed in Table 1.3.

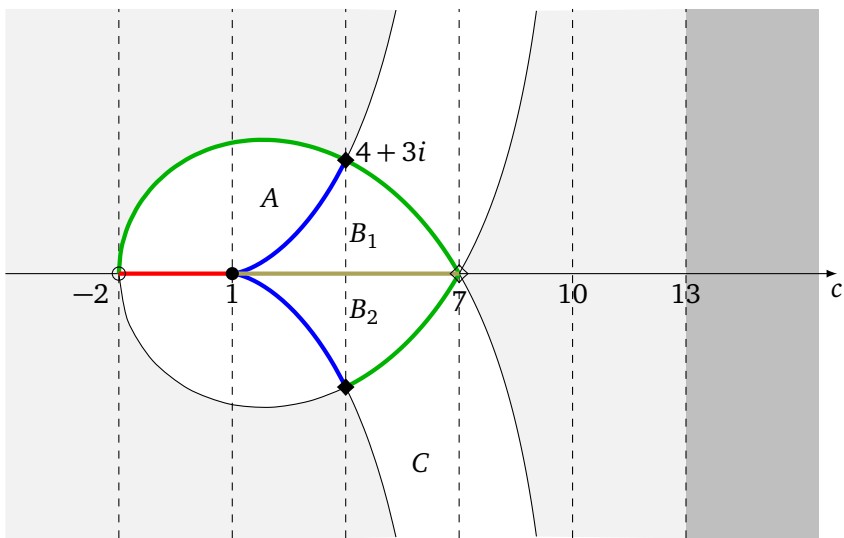

Figure 1.2: The complex $c$-plane, including the image of the complex $Q$-plane (white) via Eq. (2).

Table 1.3: Some special values of the parameters $Q$, $\beta^2$ and $c$, as related by Eq. (2) and plotted in Figures 1.1 and 1.2.

| name | value | $\beta^2$ | $c$ |
|------|-------|-----------|-----|
| $Q_1$ | $-4\sinh^2\frac{\pi}{2}\sqrt{3} \simeq -228.77$ | $e^{\pm i\frac{\pi}{3}}$ | $7$ |
| $Q_2$ | $-4\sinh^2\frac{\pi}{2} \simeq -21.18$ | $\frac{1\pm i}{2}$ | $4\pm 3i$ |
| | $0$ | $\frac{1}{2}$ | $-2$ |
| | $4$ | $1$ | $1$ |
| $Q_3$ | $4\cosh^2\pi \simeq 537.49$ | $1\pm i$ | $4\pm 3i$ |

## 2 The odd CFT, and the spectrum of the Potts model

The CFT that we will use for predicting connectivities is called the odd CFT not because it is strange, but because there is an even CFT with analogous properties [11]. The even CFT has the spectrum $\mathcal{S}_{\mathbb{Z}+\frac{1}{2},2\mathbb{Z}}$, and the two CFTs are related by analytic continuation around $c = 1$. Both CFTs can be understood in several ways:

- as limits of D-series minimal models,

- as CFTs that interpolates between D-series minimal models,

- as non-diagonal extensions of Liouville theory.

For $Q \in (0,4)$, it is the odd CFT that is expected to be related to the Potts model.

In this section we will review the odd CFT's spectrum, correlation functions, and analytic behaviour as a function of the central charge. We will then review the spectrum $\mathcal{S}^{\text{Potts}}$ [8], and its relevance to four-point connectivities [12]. And we will compare $\mathcal{S}^{\text{Potts}}$ with the spectrum $\mathcal{S}_{2\mathbb{Z},\mathbb{Z}+\frac{1}{2}}$ of the odd CFT.

### 2.1 Spectrum and correlation functions of the odd CFT

Let us describe the spectrum of the odd CFT in terms of the corresponding primary fields. We will characterize primary fields by their left and right momentums, where the momentum $P$ is related to the conformal dimension $\Delta$ by

$$\Delta(P) = \frac{c-1}{24} + P^2 \, . \tag{4}$$

We are particularly interested in momentums and dimensions of the type

$$P_{(r,s)} = \frac{1}{2}\left(r\beta - \frac{s}{\beta}\right) \quad , \quad \Delta_{(r,s)} = \Delta(P_{(r,s)}) \, , \tag{5}$$

where the numbers $r,s$ are positive integers for degenerate fields, but will take more general half-integer values. The parameter $\beta$ is related to the central charge $c$ via Eq. (2). The odd CFT has diagonal primary fields with arbitrary complex momentums, which we denote as $V_P^D$. It also has non-diagonal primary fields $V_{(r,s)}^N$ with the left and right momentums $(P_{(r,s)}, P_{(r,-s)})$, where $(r,s) \in 2\mathbb{Z} \times (\mathbb{Z} + \frac{1}{2})$. This is why the non-diagonal sector of the spectrum is called $\mathcal{S}_{2\mathbb{Z},\mathbb{Z}+\frac{1}{2}}$.

The restriction $\Re\beta^2 > 0 \iff \Re c < 13$ on the allowed central charges comes from the requirement that conformal dimensions be bounded from below (in real part), which is necessary for operator product expansions to converge. The total conformal dimension of the field $V_{(r,s)}^N$ is

$$\Delta_{(r,s)}^{\text{total}} = \Delta_{(r,s)} + \Delta_{(r,-s)} = \frac{c-1}{12} + \frac{1}{2}\left(r^2\beta^2 + s^2\beta^{-2}\right) \, , \tag{6}$$

so that $\lim_{r,s\to\infty} \Re\Delta_{(r,s)}^{\text{total}} = +\infty \iff \Re\beta^2 > 0$. This reasoning holds not only for the spectrum $\mathcal{S}_{2\mathbb{Z},\mathbb{Z}+\frac{1}{2}}$, but also for any spectrum that includes non-diagonal fields with real, unbounded values of $r,s$. This includes $\mathcal{S}^{\text{Potts}}$, as we will see in Section 2.3. Therefore, we expect that the Potts model can be analytically continued to the whole half-plane $\{\Re c < 13\}$ where the odd CFT is defined – or rather, to the double cover of that half-plane, as the spectrum depends on $\beta$ while the central charge is invariant under $\beta \to \beta^{-1}$.

The ground state of the non-diagonal sector, i.e. the state with the lowest conformal dimension, corresponds to the field $V^N_{(0,\frac{1}{2})}$. There is also a diagonal field $V^D_{P_{(0,\frac{1}{2})}}$ with the same left and right dimensions: these two fields however differ by their OPEs with other fields [13]. In order to describe cluster connectivities, we only need correlation functions of the fields $V^D_{P_{(0,\frac{1}{2})}}$ and $V^N_{(0,\frac{1}{2})}$. We will use simplified notations for these two fields,

$$V^D = V^D_{P_{(0,\frac{1}{2})}} \quad , \quad V^N = V^N_{(0,\frac{1}{2})} . \tag{7}$$

Correlation functions of these fields are invariant under the $\mathbb{Z}_2$ symmetry

$$(V^D, V^N) \to (V^D, -V^N) . \tag{8}$$

We normalize the two-point functions as

$$\left\langle V^D V^D \right\rangle = \left\langle V^N V^N \right\rangle = 1 \quad , \quad \left\langle V^D V^N \right\rangle = 0 , \tag{9}$$

where we omit the dependence on the fields' positions, which is dictated by conformal symmetry. The three-point functions are of the type

$$\left\langle V^D V^D V^D \right\rangle = -\left\langle V^D V^N V^N \right\rangle = \sqrt{D_{(0,\frac{1}{2})}} , \tag{10}$$

$$\left\langle V^D V^D V^N \right\rangle = \left\langle V^N V^N V^N \right\rangle = 0 , \tag{11}$$

where we write the three-point structure constant $\sqrt{D_{(0,\frac{1}{2})}}$ as the square-root of a four-point structure constant. The two three-point functions in the first equation must differ by a sign, and this sign is not known: we anticipate that it must be a minus sign, as we will see in a the case $c = \frac{4}{5}$ in Section 4.5.

Four-point functions that involve our two fields can be nonzero only if the number of copies of $V^N$ is even. We will not consider the four-point function

$$R_0 = \left\langle V^D V^D V^D V^D \right\rangle = \left\langle V^N V^N V^N V^N \right\rangle , \tag{12}$$

which does not appear to be related to connectivities. This leaves us with the three independent four-point functions

$$R_1 = \left\langle V^D V^D V^N V^N \right\rangle , \tag{13}$$

$$R_2 = \left\langle V^N V^D V^N V^D \right\rangle , \tag{14}$$

$$R_3 = \left\langle V^N V^D V^D V^N \right\rangle . \tag{15}$$

We will compute such four-point functions by using their $s$-, $t$- or $u$-channel expansions. For each four-point function, one expansion involves diagonal states, while the other two expansions involve non-diagonal states. Understanding the diagonal expansions is still an open problem, and we will restrict to the non-diagonal expansions. It is known that the non-diagonal expansions involve sums over the whole non-diagonal sector $\mathcal{S}_{2\mathbb{Z},\mathbb{Z}+\frac{1}{2}}$, and the corresponding four-point structure constants are also known. In particular, let us write the $t$-channel expansion of $R_1$, and the $s$-channel expansions of $R_2$ and $R_3$,

$$R_1 = \frac{1}{2} \sum_{r \in 2\mathbb{Z}} \sum_{s \in \mathbb{Z}+\frac{1}{2}} (-1)^{rs} D_{(r,s)} \mathcal{F}^{(t)}_{\Delta_{(r,s)}}(z) \mathcal{F}^{(t)}_{\Delta_{(r,-s)}}(\bar{z}) , \tag{16}$$

$$R_2 = \frac{1}{2} \sum_{r \in 2\mathbb{Z}} \sum_{s \in \mathbb{Z}+\frac{1}{2}} D_{(r,s)} \mathcal{F}^{(s)}_{\Delta_{(r,s)}}(z) \mathcal{F}^{(s)}_{\Delta_{(r,-s)}}(\bar{z}) , \tag{17}$$

$$R_3 = \frac{1}{2} \sum_{r \in 2\mathbb{Z}} \sum_{s \in \mathbb{Z}+\frac{1}{2}} (-1)^{rs} D_{(r,s)} \mathcal{F}^{(s)}_{\Delta_{(r,s)}}(z) \mathcal{F}^{(s)}_{\Delta_{(r,-s)}}(\bar{z}) , \tag{18}$$

where $z$ is the cross-ratio (45) of the four fields' positions. Due to the invariance of correlation functions under field permutations, the same structure constants $D_{(r,s)}$ appear in our three expansions. In particular, the sign factor $(-1)^{rs}$ that distinguishes $R_2$ from $R_3$ involves the conformal spin $\Delta_{(r,s)} - \Delta_{(r,-s)} = -rs$ of the $s$-channel field $V_{(r,s)}^N$, and comes from the universal behaviour of structure constants under permutations [15].

Assembling formulas for two- and three-point structure constants [13], we find the explicit expression of the four-point structure constants

$$D_{(r,s)} = \beta^{\frac{2}{\beta^2} - 2\beta^2} (-1)^{rs} \frac{\Gamma(\beta^2)\Gamma(\frac{1}{\beta^2})}{\Gamma(2-\beta^2)\Gamma(2-\frac{1}{\beta^2})} \frac{\prod_{\pm} \Gamma_\beta(\beta \pm 2P_{(r,s)})\Gamma_\beta(\frac{1}{\beta} \pm 2P_{(r,-s)})}{\Upsilon_\beta(\frac{1}{2\beta})^2 \Upsilon_\beta(\frac{1}{\beta})^2 \Upsilon_\beta(\frac{3}{2\beta})^2}$$
$$\times \prod_{\pm} \Upsilon_\beta(\tfrac{\beta}{2} + P_{(r,\pm s)})\Upsilon_\beta(\tfrac{\beta}{2} + \tfrac{1}{2\beta} + P_{(r,\pm s)})^2 \Upsilon_\beta(\tfrac{\beta}{2} + \tfrac{1}{\beta} + P_{(r,\pm s)}), \quad (19)$$

which obeys $D_{(r,s)} = D_{(r,-s)}$ although this is not completely manifest. The special functions that appear are the double Gamma function $\Gamma_\beta$, and the related Upsilon function $\Upsilon_\beta(x) = \frac{1}{\Gamma_\beta(x)\Gamma_\beta(\beta + \frac{1}{\beta} - x)}$. (The structure constants of the odd CFT are related to structure constants of Liouville theory [13].) Crossing symmetry determines these four-point structure constants only up to field renormalizations, and up to an overall field-independent normalization. We have adopted normalizations that agree with minimal models when $\beta^2$ takes appropriate rational values: normalizations such that two-point functions are one, and such that $\left\langle V_{P_{(1,1)}}^D V_P^D V_P^D \right\rangle = 1$. These normalizations underlie the expression (43) of the three-point connectivity in terms of the structure constant $D_{(0,\frac{1}{2})}$, which we compute as a special case of $D_{(r,s)}$,

$$D_{(0,\frac{1}{2})} = 4\pi^2 \beta^{\frac{2}{\beta^2} - 2\beta^2} \frac{\Gamma(\beta^2)\Gamma(\frac{1}{\beta^2})}{\Gamma(2-\beta^2)\Gamma(2-\frac{1}{\beta^2})}$$
$$\times \Gamma_\beta\left(\beta + \tfrac{1}{2\beta}\right)^3 \Gamma_\beta\left(\beta - \tfrac{1}{2\beta}\right)^3 \Gamma_\beta\left(\tfrac{1}{2\beta}\right)^3 \Gamma_\beta\left(\tfrac{3}{2\beta}\right)^3 \Upsilon_\beta\left(\tfrac{\beta}{2} - \tfrac{1}{4\beta}\right)^2 \Upsilon_\beta\left(\tfrac{\beta}{2} + \tfrac{1}{4\beta}\right)^6. \quad (20)$$

Let us briefly turn our attention to the $s$-channel conformal blocks $\mathcal{F}_\Delta^{(s)}(z)$. (The $t$-channel conformal blocks are simply $\mathcal{F}_\Delta^{(t)}(z) = \mathcal{F}_\Delta^{(s)}(1-z)$.) Virasoro conformal blocks are universal functions and are in principle known. In general they have poles at $\Delta = \Delta(P_{(r,s)})$ for $r,s \in \mathbb{N}^*$. However, our blocks are based on four fields with identical dimensions $\Delta(P_{(0,\frac{1}{2})})$, and the poles with $r \in 2\mathbb{N}+1$ have vanishing residues [9]. Moreover, our blocks have expansions of the form

$$\mathcal{F}_\Delta^{(s)}(z) = h_\infty(z)(16q(z))^\Delta \left(1 + \sum_{N \overset{2}{=} 2}^\infty c_N(\Delta)q(z)^N\right), \quad (21)$$

where the prefactor $h_\infty(z)$ and the nome $q(z)$ are known, $\Delta$-independent functions. For a term in this expansion, the even integer $N$ is called the level, and $\Delta + N$ the conformal dimension.

Expanding each conformal block in the $s$-channel decomposition (17) of the four-point function $R_2$, we obtain what we will call the $s$-channel expansion of $R_2$. A term of this expansion has the total dimension $\Delta^{\text{total}} = \Delta + N_L + \bar{\Delta} + \bar{N}$, i.e. the sum of the left and right dimensions. The terms decrease exponentially with their total dimensions, and the $s$-channel expansion converges thanks to $\lim_{r,s\to\infty} \Re\Delta_{(r,s)}^{\text{total}} = +\infty$ for $\Re\beta^2 > 0$. In order to numerically compute $R_2$, we in principle truncate the expansion to a given total dimension $\Delta^{\text{total}}$. In practice, we separately truncate $\Delta + \bar{\Delta} \leq \Delta^{\text{max}}$ and $N, \bar{N} \leq N^{\text{max}}$.

## 2.2 Behaviour at rational central charges

At rational values of $\beta^2$, both the structure constants and the conformal blocks of the odd CFT can have poles, and our four-point functions can therefore be singular.

More specifically, singularities can occur whenever the dimension $\Delta_{(r,s)}$ of a state in $\mathcal{S}_{2\mathbb{Z},\mathbb{Z}+\frac{1}{2}}$ coincides with the dimension $\Delta_{(m,n)}$ with $(m,n) \in (2\mathbb{N}^*, \mathbb{N}^*)$ of a pole of our conformal blocks. Using Eq. (5), we find

$$\Delta_{(r,s)} = \Delta_{(m,n)} \quad \Longrightarrow \quad \beta^2 \in \left\{ \frac{s-n}{r-m}, \frac{s+n}{r+m} \right\} . \tag{22}$$

Writing $\beta^2 = \frac{p}{q}$ with $p < q$ two coprime integers, singularities can therefore occur only if $q \equiv 0 \bmod 4$.

In particular, our four-point functions have finite limits as $\beta^2 \to \frac{p}{q}$ with $q \equiv 2 \bmod 4$. If moreover $p > 1$, there exists a D-series minimal model with this value of $\beta^2$, and this model has two fields with the same dimensions $\Delta = \bar{\Delta} = \Delta_{(0,\frac{1}{2})}$ as $V^D, V^N$. And our four-point functions reduce to four-point functions of that minimal model as $\beta^2 \to \frac{p}{q}$ [11]. Using the known properties of minimal models, we can deduce qualitative properties of our four-point function. In particular, if $p \neq q - 1$, the minimal model is not unitary, and some fields in its diagonal sector have negative conformal dimensions. Now the $u$-channel decomposition of our four-point function is obtained by performing the OPE of the two diagonal fields of the type $V^D$. This OPE involves all the fields in the diagonal sector, including fields with negative conformal dimensions.

In the rest of this section we focus on the limits $\beta^2 \to \frac{p}{q}$ with $q \equiv 0 \bmod 4$, where singularities do occur. To be definite we focus on the $s$-channel expansion of $R_2$. When truncated to any finite order, $R_2$ has finitely many poles of this type. But when summing over all terms, we obtain poles that are dense in the half-line $\beta^2 \in (0, \infty)$. Therefore, we should in principle not speak of poles, but rather of an essential singularity of $R_2$ on the whole half-line.

Let us discuss these poles in more detail, and determine which $s$-channel truncations are free of poles. According to [11], the behaviour of the structure constant $D_{(r,s)}$ depends on the position of $(r,s)$ with respect to the Kac table $(-\frac{q}{2}, \frac{q}{2}) \times (-\frac{p}{2}, \frac{p}{2})$ of the $(p,q)$ minimal model. As $\beta^2 \to \frac{p}{q}$, the structure constant has

- a finite limit if $(r,s)$ is in the Kac table,

- a simple pole if $(r,s)$ is on the boundary of the Kac table,

- a pole of order 2 or more otherwise.

Conformal blocks have poles too, although their leading terms are always finite. If $(r,s)$ is in the Kac table, the conformal block $\mathcal{F}^{(s)}_{\Delta_{(r,s)}}$ has simple poles at the orders $(\frac{q}{2} - r)(\frac{p}{2} - s)$ and $(\frac{q}{2} + r)(\frac{p}{2} + s)$, and other poles at higher orders. There are cancellations of poles when we combine terms in the $s$-channel expansion, and in particular we expect all poles of order 2 or more to cancel, leaving at most simple poles in any truncation of our four-point function [11]. The terms from the boundary of the Kac table have simple poles that cannot be cancelled by other terms. The lowest-dimensional pole of this type appears at level $(0,0)$ in the term $D_{(0,\frac{p}{2})} \mathcal{F}^{(s)}_{\Delta_{(0,\frac{p}{2})}}(z) \mathcal{F}^{(s)}_{\Delta_{(0,\frac{p}{2})}}(\bar{z})$, whose structure constant $D_{(0,\frac{p}{2})}$ has a simple pole. The total conformal dimension of this term is

$$\Delta^{\min}_{p,q} = 2\Delta_{(0,\frac{p}{2})} = \frac{c-1}{12} + \frac{pq}{8} . \tag{23}$$

For $(r,s)$ in the Kac table, the term $D_{(r,s)}\mathcal{F}^{(s)}_{\Delta_{(r,s)}}(z)\mathcal{F}^{(s)}_{\Delta_{(r,-s)}}(\bar{z})$ also has an uncancellable simple pole at the level $((\frac{q}{2}+r)(\frac{p}{2}+s),0)$. The total dimension of this pole is

$$\Delta_{(q-r,s)} + \Delta_{(r,-s)} = \Delta^{\min}_{p,q} + \frac{1}{2pq}\left(pr+qs-\tfrac{1}{2}pq\right)^2 . \tag{24}$$

This is larger than $\Delta^{\min}_{p,q}$, which is therefore the lowest total dimension of a simple pole. Any truncation below the dimension $\Delta^{\min}_{p,q}$ is therefore regular as $\beta^2 \to \frac{p}{q}$.

Let us focus on the interval $1 \leq Q \leq 4$, where we did our Monte-Carlo calculations. This corresponds to $\frac{2}{3} \leq \beta^2 \leq 1$. In this interval, the rational values of $\beta^2 = \frac{p}{q}$ with $q \equiv 0 \bmod 4$ and $p < q$ are, in order of increasing $q$,

$$\beta^2 = \frac{3}{4}, \frac{7}{8}, \frac{11}{12}, \frac{11}{16}, \frac{13}{16}, \frac{15}{16}, \cdots . \tag{25}$$

For $\beta^2 = \frac{3}{4}$, we have a rather low $\Delta^{\min}_{3,4} \sim 1.46$, so no reasonable truncation can eliminate all the poles. However, in all other cases, we have $\Delta^{\min}_{p,q} > 6.91$, and there exist truncations that eliminate all the poles while keeping many terms in the $s$-channel expansion. For example, for $\beta^2 = \frac{7}{8}$, the lowest-dimensional pole from Kac table fields comes from a level 4 null vector of $V^N_{(2,\frac{3}{2})}$, since $\Delta_{(2,\frac{3}{2})} = \Delta_{(2,2)}$.

## 2.3 The spectrum of the Potts model

Let us discuss the spectrum of [12], and compare it with our spectrum $\mathcal{S}_{2\mathbb{Z},\mathbb{Z}+\frac{1}{2}}$. In [12], Jacobsen and Saleur do not content themselves with the partition function: they study four-point connectivities and conjecture their spectrums in each channel. However, their conjectures are formulated at the level of conformal dimensions, i.e. eigenvalues of the Virasoro generator $L_0$, and do not capture the full action of the Virasoro algebra. This information is potentially within reach of their lattice algebraic methods, or of CFT-motivated guesses as in [9] (Appendix A.3). This issue will however not affect our comparison, because for generic $Q$ our spectrum $\mathcal{S}_{2\mathbb{Z},\mathbb{Z}+\frac{1}{2}}$ is made of irreducible Verma modules. The structure of an irreducible Verma module is entirely determined by its conformal dimension; ambiguities can only arise in degenerate representations, whose conformal dimensions are of the form $\Delta_{(r,s)}$ with $r,s \in \mathbb{N}^*$.

We call the four independent four-point connectivities of the critical 2d Potts model $P_0, P_1, P_2, P_3$, see Eq. (44) for their precise definitions. For the moment we only need to know that these connectivities have the same behaviour under permutations as the correlation functions $R_\sigma$ (12)-(15) of the odd CFT, in particular $P_0$ is invariant under all permutations of the four points. Let us display the $s$-channel spectrums of these connectivities. The $s$-channel spectrum of $P_\sigma$ is the set of exponents $\Delta, \bar{\Delta}$ such that

$$P_\sigma \underset{z \to 0}{=} \sum_{(\Delta,\bar{\Delta})\in\mathcal{S}} D_{\Delta,\bar{\Delta}} z^{\Delta-2\Delta_{(0,\frac{1}{2})}} \bar{z}^{\bar{\Delta}-2\Delta_{(0,\frac{1}{2})}}\left(1+O(z)\right), \tag{26}$$

for some $z$-independent coefficients $D_{\Delta,\bar{\Delta}}$. Since $R_\sigma$ is a four-point function in a consistent CFT, its $s$-channel spectrum controls its decomposition into conformal blocks (for example Eq. (16)). The $s$-channel spectrum of $P_\sigma$ is given in the weaker sense of controlling its asymptotic properties, but we do not know whether the entire series $\left(1+O(z)\right)$ are Virasoro conformal blocks, and we do not know whether the coefficients $D_{\Delta,\bar{\Delta}}$ are squares of three-point structure constants. In any case, the difference $\Delta - \bar{\Delta}$ between the left and right dimensions is called the conformal spin, and it should be integer for $P_\sigma$ to have no monodromy as $z_2$ goes around

$z_1$. The spectrums of $P_\sigma$ are [12]

| connectivity | $s$-channel spectrum | lowest states | |
|---|---|---|---|
| $P_0$ | $\mathcal{S}_{0,\mathbb{Z}+\frac{1}{2}} \cup \mathcal{S}_{2\mathbb{Z}^*,\mathbb{Q}}$ | $(0,\frac{1}{2}),(2,0),(0,\frac{3}{2}),(2,1)$ | (27) |
| $P_1$ | $\mathcal{S}^D_{1,\mathbb{N}^*} \cup \mathcal{S}_{0,\mathbb{Z}+\frac{1}{2}} \cup \mathcal{S}_{2\mathbb{Z}^*,\mathbb{Q}}$ | $(1,1)^D,(0,\frac{1}{2}),(1,2)^D,(2,0)$ | |
| $P_2,P_3$ | $\mathcal{S}_{2\mathbb{Z}^*,\frac{1}{2}\mathbb{Q}}$ | $(2,0),(2,\frac{1}{2}),(2,1),(2,\frac{3}{2})$ | |

where we introduced the notations

$$\mathcal{S}_{2\mathbb{Z}^*,\mathbb{Q}} = \left\{(\Delta_{(r,s)},\Delta_{(r,-s)})\right\}_{r\in 2\mathbb{Z}^*,s\in\mathbb{Q},rs\in 2\mathbb{Z}} \quad , \quad \mathcal{S}^D_{1,\mathbb{N}^*} = \left\{(\Delta_{(1,s)},\Delta_{(1,s)})\right\}_{s\in\mathbb{N}^*} . \tag{28}$$

(In the first version of [12] there was $\mathcal{S}^D_{1,\mathbb{Z}}$ instead of $\mathcal{S}^D_{1,\mathbb{N}^*}$, but we never observed the states $(1,s)^D$ with $s \leq 0$, and the authors of [12] now state that they are absent.) The union of all these $s$-channel spectrums is supposed to be a subset of the total spectrum $\mathcal{S}^{\text{Potts}}$. The energy operator appears in the sector $\mathcal{S}^D_{1,\mathbb{N}^*}$ with the indices $(1,2)$. For $Q = 1$, the energy operator appeared in the Coulomb gas analysis of [16] and also in the bootstrap analysis of [17]. For each spectrum we indicate the few states with the lowest total conformal dimension, assuming $Q \in (0,4)$ i.e. $\frac{1}{2} < \beta^2 < 1$, see Figure 2.1.

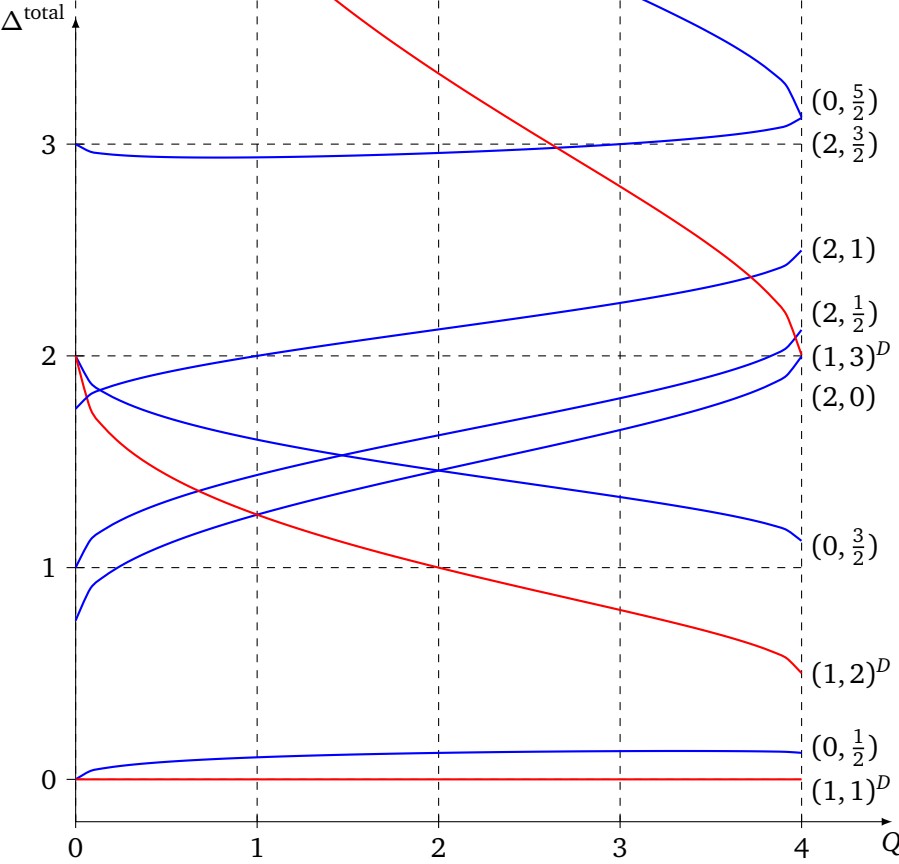

Figure 2.1: Conformal dimensions of low-lying states in the Potts model as functions of $Q$. We use red for $\mathcal{S}^D_{1,\mathbb{N}^*}$ and blue for $\mathcal{S}_{0,\mathbb{Z}+\frac{1}{2}} \cup \mathcal{S}_{2\mathbb{Z}^*,\mathbb{Q}}$.

A particularly clean comparison between connectivities and four-point functions can be done in the $s$-channel expansion of $R_2 - R_3$, whose spectrum is relatively sparse. According to the expansions (17)-(18) of the two relevant four-point functions, the $s$-channel spectrum

of $R_2 - R_3$ is indeed the odd spin sector of $\mathcal{S}_{2\mathbb{Z}, \mathbb{Z}+\frac{1}{2}}$, namely $\mathcal{S}_{4\mathbb{Z}+2, \mathbb{Z}+\frac{1}{2}}$. The corresponding combination of connectivities is $P_2 - P_3$, and its $s$-channel spectrum is the odd spin sector of $\mathcal{S}_{2\mathbb{Z}^*, \frac{1}{2}\mathbb{Q}}$, namely

$$\mathcal{S}^{\text{odd spin}}_{2\mathbb{Z}^*, \frac{1}{2}\mathbb{Q}} = \mathcal{S}_{4\mathbb{Z}+2, \mathbb{Z}+\frac{1}{2}} \cup \mathcal{S}_{8\mathbb{Z}+4, \mathbb{Z}+\frac{1}{4}} \cup \mathcal{S}_{12\mathbb{Z}+6, \mathbb{Z}+\frac{1}{6}} \cup \mathcal{S}_{16\mathbb{Z}+8, \mathbb{Z}+\frac{1}{8}} \cup \mathcal{S}_{16\mathbb{Z}+8, \mathbb{Z}+\frac{3}{8}} \cup \cdots . \tag{29}$$

This spectrum contains many states beyond our odd spin spectrum $\mathcal{S}_{4\mathbb{Z}+2, \mathbb{Z}+\frac{1}{2}}$. However, the states with the lowest total conformal dimensions do belong to $\mathcal{S}_{4\mathbb{Z}+2, \mathbb{Z}+\frac{1}{2}}$. Therefore, $\mathcal{S}_{4\mathbb{Z}+2, \mathbb{Z}+\frac{1}{2}}$ can be seen as an approximation of $\mathcal{S}^{\text{odd spin}}_{2\mathbb{Z}^*, \frac{1}{2}\mathbb{Q}}$. Assuming that the Potts model's exact spectrum (in this channel) is indeed $\mathcal{S}^{\text{odd spin}}_{2\mathbb{Z}^*, \frac{1}{2}\mathbb{Q}}$, this suggests that the odd CFT provides an approximation of the connectivity that corresponds to $R_2 - R_3$.

To get an idea of how good an approximation the odd CFT provides, let us plot the conformal dimensions $\Delta^{\text{total}} = \Delta + \bar{\Delta}$ of the first few states in the odd spin sector of $\mathcal{S}_{2\mathbb{Z}^*, \mathbb{Q}}$. The state with the lowest dimension that is not in $\mathcal{S}_{4\mathbb{Z}+2, \mathbb{Z}+\frac{1}{2}}$ has the indices $(r, s) = (4, \frac{1}{4})$, and for $Q \in (0, 4)$ it is at best the third-lowest state in the spectrum, see Fig. 2.2.

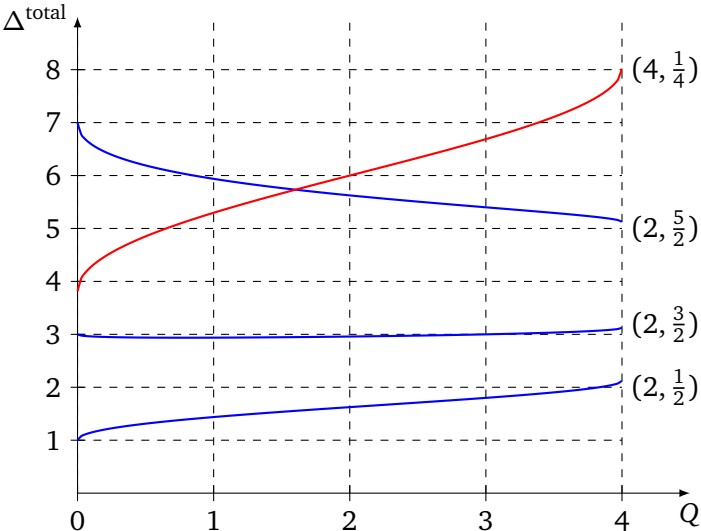

Figure 2.2: Conformal dimensions of low-lying states in the odd spin sector of the Potts model, as functions of $Q$. We use blue for states that belong to the odd CFT, and red for the state $(4, \frac{1}{4})$ that does not.

This comparison of conformal dimensions actually understates how good an approximation the odd CFT provides, because the structure constant for the state $(4, \frac{1}{4})$ is very small for most values of $Q$ [12]. As we will see in Section 4, the odd CFT is expected to exactly describe connectivities for $Q = 0, 3, 4$, so the structure constant $D_{(4, \frac{1}{4})}$ must vanish at these values of $Q$. We therefore expect the approximation to be very good near $Q = 0, 3, 4$, and to become less good near the rational values of $\beta^2$ that were discussed in Section 2.2. Moreover, the approximation is very good near $z = 0$ where the $s$-channel expansion is dominated by the lowest-lying states, and becomes less good near $z = 1, \infty$ where the expansion diverges.

## 3  Monte-Carlo computation of connectivities

### 3.1  Definition and basic properties of connectivities

Define a graph $\mathcal{G}$ as a collection of bonds between neighbouring sites of a lattice. Here we will always consider the square lattice $\mathbb{Z}^2$. Each edge of the lattice either has a bond, or no bond. According to these bonds, the lattice is split into a disjoint union of connected clusters. We consider here the random cluster $Q$-state Potts model [1] defined by the partition function

$$\mathcal{Z} = \sum_{\mathcal{G}} \text{Probability}(\mathcal{G}) \,, \tag{30}$$

where the probability of a graph is defined as

$$\text{Probability}(\mathcal{G}) = Q^{\# \text{ clusters}} p^{\# \text{ bonds}} (1-p)^{\# \text{ edges without bond}} \,. \tag{31}$$

The bond probability has a critical value

$$p_c = \frac{\sqrt{Q}}{\sqrt{Q}+1} \,, \quad \text{(Square lattice)} \,, \tag{32}$$

where the probability that there exists a percolating cluster jumps from 0 to 1, in the limit of infinite lattice size. In order to obtain a conformally invariant model, we set the bond probability to its critical values, send the lattice size $L$ to infinity, and study objects at a scale $r$ such that $1 \ll r \ll L$, where both $r$ and $L$ are expressed in units of the lattice spacing. This is called taking the scaling limit:

$$\text{Scaling limit:} \quad p = p_c \,, \quad 1 \ll r \ll L \,. \tag{33}$$

In the Potts model, connectivities are defined as probabilities that a number of lattice points belong to the same or different clusters. The number of independent $n$-point connectivities is given in Eq. (1). We are interested in the two- and three-point connectivities,

$$p_{12} = \text{Probability}(z_1, z_2 \text{ are in the same cluster}) \,, \tag{34}$$
$$p_{123} = \text{Probability}(z_1, z_2, z_3 \text{ are all in the same cluster}) \,, \tag{35}$$

and in the four independent four-point connectivities,

$$p_{1234} = \text{Probability}(z_1, z_2, z_3, z_4 \text{ are all in the same cluster}) \,, \tag{36}$$
$$p_{12,34} = \text{Probability}(z_1, z_2 \text{ and } z_3, z_4 \text{ are in two different clusters}) \,, \tag{37}$$
$$p_{13,24} = \text{Probability}(z_1, z_3 \text{ and } z_2, z_4 \text{ are in two different clusters}) \,, \tag{38}$$
$$p_{14,23} = \text{Probability}(z_1, z_4 \text{ and } z_2, z_3 \text{ are in two different clusters}) \,. \tag{39}$$

Coulomb gas arguments show that the scaling limits of connectivities behave as correlation functions of primary operators with the total conformal dimension $2\Delta_{(0,\frac{1}{2})}$ [18]. This dimension is given as a function of $Q$ by Eq. (5) together with Eq. (2), see Figure (2.1) for a plot. In order to take the scaling limit, we assume that the positions $z_i$ are of order $r$. Then our connectivities go to zero as $r \to \infty$. We define finite, nontrivial limits of the type

$$P_{12\cdots n}\left(\tfrac{z_1}{r}, \tfrac{z_2}{r}, \cdots, \tfrac{z_n}{r}\right) = \lim_{\substack{1 \ll r \ll L \\ \frac{z_i}{r} \text{ fixed}}} a_0^{-\frac{n}{2}} r^{2n\Delta_{(0,\frac{1}{2})}} p_{12\cdots n}(z_1, z_2, \cdots, z_n) \,, \tag{40}$$

where we will shortly determine the number $a_0$ by normalizing the two-point connectivity. Due to the prefactor $a_0^{-\frac{n}{2}} r^{2n\Delta_{(0,\frac{1}{2})}}$, this limit is no longer a probability, and in particular it is not necessarily less than one.

Conformal invariance determines the two- and three-point connectivities up to a constant prefactor,

$$P_{12} = \left| \frac{z_{12}}{r} \right|^{-4\Delta_{(0,\frac{1}{2})}} , \tag{41}$$

$$P_{123} = C \left| \frac{z_{12}}{r} \right|^{-2\Delta_{(0,\frac{1}{2})}} \left| \frac{z_{13}}{r} \right|^{-2\Delta_{(0,\frac{1}{2})}} \left| \frac{z_{23}}{r} \right|^{-2\Delta_{(0,\frac{1}{2})}} , \tag{42}$$

where $z_{ij} = z_i - z_j$, and the condition that $P_{12}$ involves no numerical prefactor determines the non-universal number $a_0$ in Eq. (40), see Appendix A. The factor $C$ is a universal quantity, whose value was conjectured [6] to be

$$C = \sqrt{2D_{(0,\frac{1}{2})}} , \tag{43}$$

where $D_{(0,\frac{1}{2})}$ was given in Eq. (20). This conjecture was verified numerically in [19–21]. The factor $\sqrt{2}$ has been explained in [6] and in [21] as being related to a two-fold degeneracy in the spectrum. This degeneracy can be understood as coming from an analytic continuation in $Q$ of the permutation symmetry at $Q = 3$ (see Eq. (125) below). In [21], this degeneracy was shown to appear at the level of the transfer matrix for any value of $Q$. This conjecture has moreover been generalized to $O(n)$ loop models [21,22] and to fully packed loop models [23].

Let us now discuss the scaling limits of the four-point connectivities. We introduce the shorter notations

$$P_0 = P_{1234} , \quad P_1 = P_{12,34} , \quad P_2 = P_{13,24} , \quad P_3 = P_{14,23} . \tag{44}$$

Up to simple factors, conformal invariance reduces these quantities to functions of the cross-ratio

$$z = \frac{z_{12} z_{34}}{z_{13} z_{24}} . \tag{45}$$

Namely,

$$P_\sigma \left( \frac{z_1}{r}, \frac{z_2}{r}, \frac{z_3}{r}, \frac{z_4}{r} \right) = \left| \frac{z_{13}}{r} \frac{z_{24}}{r} \right|^{-4\Delta_{(0,\frac{1}{2})}} P_\sigma(z) . \tag{46}$$

(We may write $P_\sigma$ for $P_\sigma \left( \frac{z_1}{r}, \frac{z_2}{r}, \frac{z_3}{r}, \frac{z_4}{r} \right)$ or $P_\sigma(z)$, depending on the context.) The dimensions of the $s$-channel states are apparent in the limit $z \to 0$ (26), equivalently $\frac{z_{12}}{r} \to 0$ and/or $\frac{z_{34}}{r} \to 0$.

Let us sketch how scaling limits of connectivities behave for $z \to 0$. Our arguments will be heuristic, as we will assume without proof that we can exchange the scaling limit with the limit $z \to 0$. In order to demonstrate this type of arguments, let us compare the $\frac{z_{12}}{r} \to 0$ limits of the two- and three-point connectivities. We write $P_{123} = P_{12|23} P_{23}$, where $P_{12|23}$ is a conditional probability, and we predict $P_{12|23} \underset{\frac{z_{12}}{r} \to 0}{\ll} P_{12}$: if $z_2$ is already known to belong to a large cluster that includes $z_3$, then the probability that $z_1$ also belongs to that cluster depends less sharply on its distance to $z_2$. We deduce $P_{123} \underset{\frac{z_{12}}{r} \to 0}{\ll} P_{12}$, which agrees with Eq. (41)-(42). On the other hand, we expect $P_{12|23} \underset{\frac{z_{12}}{r} \to 0}{\sim} P_{12|234}$: adding the point $z_4$ to the large cluster does not much change the probability that $z_1$ belongs to it, as is particularly clear if $\frac{z_{34}}{r} \to 0$. It follows that

$$P_{1234} = P_{12|234} P_{234} \underset{\frac{z_{12}}{r} \to 0}{\sim} P_{12|23} P_{234} = \frac{P_{123} P_{234}}{P_{23}} , \tag{47}$$

which implies

$$P_0(z) \underset{z \to 0}{\sim} C^2 |z|^{-2\Delta_{(0,\frac{1}{2})}} . \tag{48}$$

When it comes to $P_{12,34}$, the behaviour as $z_{12}, z_{34} \to 0$ should factorize,

$$P_{12,34} \underset{\frac{z_{12}}{r} \to 0}{\sim} P_{12} P_{34} . \tag{49}$$

And since it becomes more difficult for two points to belong to different clusters as they come closer to one another, we expect

$$P_{13,24}, P_{14,23} \underset{z \to 0}{\ll} P_{1234} . \tag{50}$$

Finally, let us discuss the subleading behaviour of $P_{12 \cap 34} = P_{1234} + P_{12,34}$. We already know the leading behaviour $P_{12 \cap 34} \underset{\frac{z_{12}}{r} \to 0}{\sim} P_{12} P_{34}$. Corrections to that behaviour must come from clusters that approach all four points, and the probability for such clusters to exist is of the order of $P_{1234}$. But in the limit $\frac{z_{12}}{r} \to 0$, most such clusters do not distinguish $z_1$ from $z_2$, and cannot contribute to the corrections. We conclude

$$P_{1234} + P_{12,34} \underset{\frac{z_{12}}{r} \to 0}{\sim} P_{12} P_{34} + o(P_{1234}) . \tag{51}$$

To summarize, the leading total dimensions that control the $z \to 0$ limits (26) of $P_\sigma$ should be

| scaling limit of connectivity | leading total dimensions |
|---|---|
| $P_0$ | $2\Delta_{(0,\frac{1}{2})}$ |
| $P_1$ | $0$ |
| $P_0 + P_1$ | $0, > 2\Delta_{(0,\frac{1}{2})}$ |
| $P_2, P_3$ | $> 2\Delta_{(0,\frac{1}{2})}$ |

$$\tag{52}$$

This agrees with the predicted spectrums (27) of [12], given $\Delta_{(1,1)} = 0$ and $\Delta_{(2,0)} > \Delta_{(0,\frac{1}{2})}$. Moreover, we have additional predictions on the first subleading correction to $P_0 + P_1$.

## 3.2 Monte-Carlo algorithm and scaling limit

In order to simulate the $Q$-state Potts random cluster model, we have implemented the Chayes-Machta Algorithm [24, 25]. This algorithm in fact reduces the problem with general $Q$ to a variant of a Swendsen-Wang algorithm used for spin systems with $[Q]$ number of states, $[Q]$ being the integer part of $Q$. The configuration space scales therefore as $[Q]^{\text{number of sites}}$.

We performed the simulations for the following values of $Q$:

$$Q = 1 + \frac{j}{4} \quad \text{for} \quad j = 0, 1, \ldots, 9 , \tag{53}$$

$$Q = 4\cos^2 \frac{7\pi}{10} , \quad 4\cos^2 \frac{7\pi}{8} = 2 + \sqrt{2} . \tag{54}$$

These include the special cases $Q = 2, 3, 4$ where exact solutions are known for spin correlation functions and/or cluster connectivities, the cases $Q = 4\cos^2 \frac{7\pi}{10}, 3$ where the odd CFT reduces to D-series minimal models, and the values $Q = 2, 2 + \sqrt{2}$ where four-point functions in the odd CFT have poles at relatively low conformal dimensions. (See Section 2.2.)

We assumed periodic boundary conditions on both directions of the lattice, thus considering the model on a torus. We collected data on lattices of various linear sizes up to

$L = 2^{13} = 8192$. In order to measure the four-point connectivities, we considered different locations for the points $z_i$, like for instance setting the points on a circle or on the vertexes of a parallelograms. We verified that (up to simple factors) the scaling limits of the connectivities depend only on the cross-ratio (45), as dictated by conformal invariance. We settled on setting the four points on the vertexes of a rectangle of size $r \times \lambda r$,

$$z_1 = ir \; ; \; z_2 = 0 \; ; \; z_3 = \lambda r \; ; \; z_4 = (i + \lambda)r \; , \tag{55}$$

so that the cross-ratio is

$$z = \frac{1}{1 + \lambda^2} \; . \tag{56}$$

With this geometry, the connectivities and their scaling limits have to obey the following inequalities,

$$P_{13,24} < P_{14,23} \quad , \quad P_2 < P_3 \; . \tag{57}$$

We used rectangles with $\lambda \in [1, 8]$ i.e. $z \in [\frac{1}{65}, \frac{1}{2}]$. This almost covers the interval $z \in [0, \frac{1}{2}]$, which is itself enough for deducing the behaviour of connectivities on the whole real line $z \in \mathbb{R}$, using their behaviour under permutations of $\{z_i\}$. (Such permutations indeed include $z \to 1 - z$ and $z \to \frac{1}{z}$.) We could compute connectivities for any $z \in \mathbb{C}$, but focussing on the real line is enough for studying the limit $z \to 0$, which gives us access to the total conformal dimensions of the $s$-channel states.

We therefore compute probabilities that depend on $L$ and $r$, in addition to the cross-ratio $z$. Let us explain how we extract the scaling limits $P_\sigma$, which depend on $z$ only. The dependence on $L, r$ manifests itself as three types of corrections to the scaling limit (see Appendix A for more details):

1. IR corrections, related to the finite size of the lattice. Such corrections are related to the presence of irrelevant operators. With our lattice sizes, they are negligible.

2. UV corrections, also known as lattice effects, due to the lattice discretization of space. We find that such corrections induce a relatively weak dependence on $r$.

3. Topological corrections, due to our space being a torus. These induce a significant dependence on $\frac{r}{L}$.

In order to take these corrections into account, we have used the ansatz

$$p_{1234}(z|r, L) = a_0^2 \frac{P_0(z)}{r^{8\Delta_{(0,\frac{1}{2})}}} \left(1 + c_1(z)\left(\frac{r}{L}\right)^{2\Delta_{(1,2)}}\right)\left(1 + \frac{b_1(z)}{r} + \frac{b_2(z)}{r^2}\right) \; . \tag{58}$$

We use this same ansatz for determining $P_1, P_2, P_3$ from $p_{12,34}, p_{13,24}, p_{14,23}$ respectively. Our ansatz parametrizes UV corrections by two coefficients $b_1, b_2$, and topological corrections by one coefficient $c_1$. The non-trivial observation is that the topological corrections are accounted for by a term with the exponent $2\Delta_{(1,2)}$, which is the total conformal dimension of the energy operator. On the other hand, the UV corrections are not universal, and depend on the specific geometry of the lattice; the choice of a function for parametrizing them is neither fundamental nor very important. As discussed in Appendix A, our numerical procedure provides the function $P_\sigma$ with up to four significant digits.

Table 3.1: Numerical determination of the coefficients $\lambda$ and $\mu$ in Eq. (59).

| $Q$ | $\lambda$ | $\mu(Q-2)$ |
|---|---|---|
| 1 | 0.500046 | 2.01989 |
| 1.25 | 0.499868 | 2.01543 |
| 1.38197 | 0.499977 | 2.01251 |
| 1.5 | 0.499939 | 2.00881 |
| 1.75 | 0.500062 | 2.00426 |
| 2.25 | 0.499968 | 2.00252 |
| 2.5 | 0.500089 | 2.00018 |
| 2.75 | 0.500364 | 1.99476 |
| 3.0 | 0.500845 | 1.98729 |

### 3.3 Comparison with the odd CFT

Let us compare the $P_\sigma$ with the predictions of the odd CFT. This comparison was already sketched in [9], where a good agreement was found. Here, we will show that the agreement still holds when we increase the precision of Monte-Carlo calculations. Moreover, we will give simple analytic formulas for the coefficients of the linear relation between the connectivities and the correlation functions, while stressing that the relation cannot be exact.

The basic ansatz for the relation between the three correlation functions $R_1, R_2, R_3$, and the four connectivities $P_0, P_1, P_2, P_3$, is

$$R_\sigma = \lambda (P_0 + \mu P_\sigma), \qquad (\sigma = 1,2,3), \tag{59}$$

where $\lambda$ and $\mu$ are $Q$-dependent coefficients. This relation was initially found as the result of numerical observations. Notice that the term $P_1 + P_2 + P_3$ is absent, although it is allowed by permutation symmetry. This is because, according to Eq. (52),

$$P_1(z) \underset{z \to 0}{\gg} R_2(z), R_3(z). \tag{60}$$

These results differ from those of [9], because we no longer use the normalization $D_{(0,\frac{1}{2})} = 1$ of that work. With our normalization, the numerically determined values of $\lambda$ and $\mu$ (see Table 3.1) suggest the simple analytic formulas $\lambda = \frac{1}{2}$ and $\mu = \frac{2}{Q-2}$. There seem to be significant deviations from these formulas in the case $Q = 3$: these deviations are however artefacts of the loss of numerical precision as $Q$ increases, and we can increase the precision by allowing a second topological correction in the ansatz (58). The suggested relation is therefore

$$R_\sigma = \frac{1}{2}P_0 + \frac{1}{Q-2}P_\sigma \quad , \qquad (\sigma = 1,2,3). \tag{61}$$

The coefficient of $P_0$ can be understood by considering the limit $z \to 0$:

$$R_2(z) \underset{z \to 0}{\sim} R_3(z) \underset{z \to 0}{\sim} D_{(0,\frac{1}{2})}|z|^{-2\Delta_{(0,\frac{1}{2})}}. \tag{62}$$

The coefficient of $P_0$ is therefore dictated by the relation (43) between the three-point connectivity $C$, and the structure constant $D_{(0,\frac{1}{2})}$. However, we do not have an explanation for the very simple form of the coefficient of $P_\sigma$.

Comparing $R_2, R_3$ with the corresponding combinations of connectivities, we find an excellent agreement, see Figure 3.2 for the case $Q = 2.5$. (Since $R_1, R_2, R_3$ are related to one another by permutations of field positions, it is in principle enough to study one of them.)

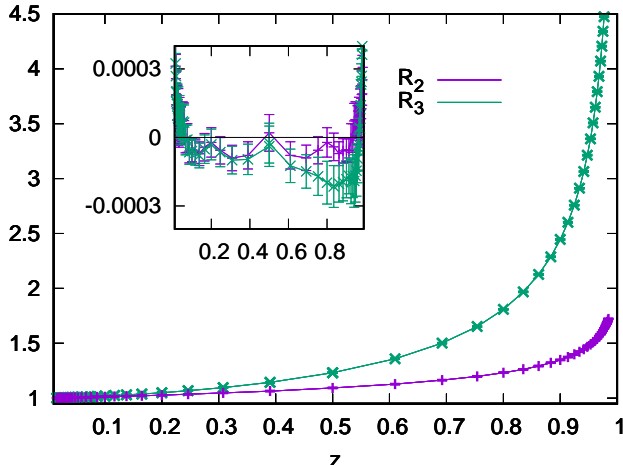

Figure 3.2: Comparing $R_2, R_3$ (lines) with the corresponding combinations of connectivities (crosses) for $Q = 2.5$. We plot these quantities as functions of $z \in (0, 1)$, and find that Eq. (61) is obeyed. The inset shows the relative differences $\frac{R_\sigma - \frac{1}{2}P_0 - \frac{1}{Q-2}P_\sigma}{R_\sigma}$, which are $O(10^{-4})$ for most values of $z$.

We stress that our analytic arguments rule out the relation (61) to be exact. Connectivities are expected to be smooth functions of $Q$, and our Monte-Carlo results do not contradict this expectation. Four-point functions of the odd CFT have simple poles, and these poles are dense in the segment $Q \in (1, 4)$, as we reviewed in Section 2.2. The pole with the lowest conformal dimension occurs at $Q = 2$ i.e. $\beta^2 = \frac{3}{4}$, but this pole is taken care of by the coefficient $\frac{1}{Q-2}$ in our relation, as we will shortly demonstrate. The next pole occurs at $Q = 2 + \sqrt{2}$ i.e. $\beta^2 = \frac{7}{8}$, where four-point functions have a narrow peak with a small residue. This pole comes from the conformal block $\mathcal{F}^{(s)}_{(2,\frac{3}{2})}$, and it is present in our calculation of the four-point function provided we truncate the block at a level $N^{\max} \geq 4$. (See Section 2.1.) However, truncating the block at a lower level, or simply ignoring the divergent term, lead to a four-point function that agrees with Monte-Carlo results. In spite of the pole, the odd CFT is therefore able to reproduce all our Monte-Carlo results down to their actual numerical precision.

Let us finally test the relation (61) near $Q = 2$, where both sides have a simple pole. We expand the correlation functions and the connectivities near the pole, temporarily using a notation that makes their dependence on $Q$ explicit:

$$R_\sigma[Q] \underset{Q \to 2}{=} \frac{1}{Q-2} R_\sigma^{\mathrm{Res}}[2] + R_\sigma^{\mathrm{Reg}}[2] + O(Q-2) \,, \tag{63}$$

$$P_\sigma[Q] \underset{Q \to 2}{=} P_\sigma[2] + (Q-2)P'_\sigma[2] + O((Q-2)^2) \,, \tag{64}$$

where $P'_\sigma[2]$ is a derivative with respect to $Q$ at $Q = 2$. Our relation predicts the two identities

$$R_\sigma^{\mathrm{Res}}[2] = P_\sigma[2] \quad , \quad R_\sigma^{\mathrm{Reg}}[2] = \frac{1}{2}P_0[2] + P'_\sigma[2] \,. \tag{65}$$

These identities are plotted in Figure 3.3 and Figure 3.4 respectively, and we find that they are obeyed to a very good accuracy. The worsening of the agreement near $z = 1$ can be attributed to the divergence of the conformal block expansion of $R_\sigma[2]$ at $z = 1$.

## 3.4 Asymptotic behaviour at coinciding points

Let us use our Monte-Carlo results for investigating the behaviour of the connectivities in the limit $z \to 0$. This limit allows us to determine the first few terms of $s$-channel conformal block

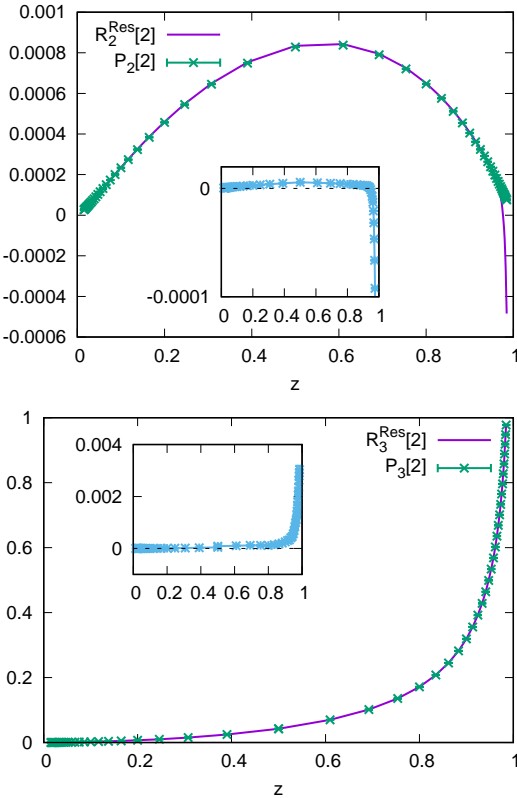

Figure 3.3: Residues of $R_2, R_3$ at $Q = 2$, compared with values of $P_2, P_3$ at $Q = 2$, as functions of $z$. The insets give the differences $R_\sigma^{\text{Res}}[2] - P_\sigma[2]$.

decompositions of the type

$$P_\sigma(z) = \sum_{\Delta, \bar{\Delta}} D_{\Delta, \bar{\Delta}} \mathcal{F}_\Delta^{(s)}(z) \mathcal{F}_{\bar{\Delta}}^{(s)}(\bar{z}) . \tag{66}$$

The $s$-channel conformal blocks $\mathcal{F}^{(s)}$ are universal quantities, and the parameters to be determined are the spectrum $\{\Delta, \bar{\Delta}\}$ and structure constants $D_{\Delta, \bar{\Delta}}$.

We do this by fitting Monte-Carlo data to truncated decompositions, see Appendix B for the details. This amounts to using ansatzes of the form

$$P_\sigma(z) \underset{z>0}{=} \delta_{\sigma,1} + \sum_{k=1}^{K} \alpha_\sigma^{(k)} z^{\beta_\sigma^{(k)}} \left( 1 + \tfrac{1}{2} \beta_\sigma^{(k)} z \right) , \tag{67}$$

where the $\beta_\sigma^{(k)}$ are meant to increase with $k$. In these ansatzes, we replace a product of conformal blocks $\mathcal{F}_\Delta^{(s)}(z) \mathcal{F}_{\bar{\Delta}}^{(s)}(\bar{z})$ with the first two terms of its series expansion near $z = 0$. In the case $\sigma = 1$ we explicitly include the contribution of the identity channel $\Delta = \bar{\Delta} = 0$, which is constant in this approximation, and more specifically 1 according to Eq. (49). In Figure 3.5, we plot the Monte-Carlo results at $Q = 1$, together with the best one-term and two-term fits. (The qualitative behaviour is the same for any $Q \in (1, 4)$.) In the limit $z \to 0$, we observe a hierarchy $P_2 < P_3 \ll P_0, P_1$, which agrees with the basic predictions of Section 3.1 and Eq. (57). On the other hand, the prediction $P_0 \ll P_1$ is not manifestly satisfied, because the displayed data do not reach close enough to $z = 0$. (But our fits for $P_0$ and $P_1$ manifestly obey $P_0 \ll P_1$.)

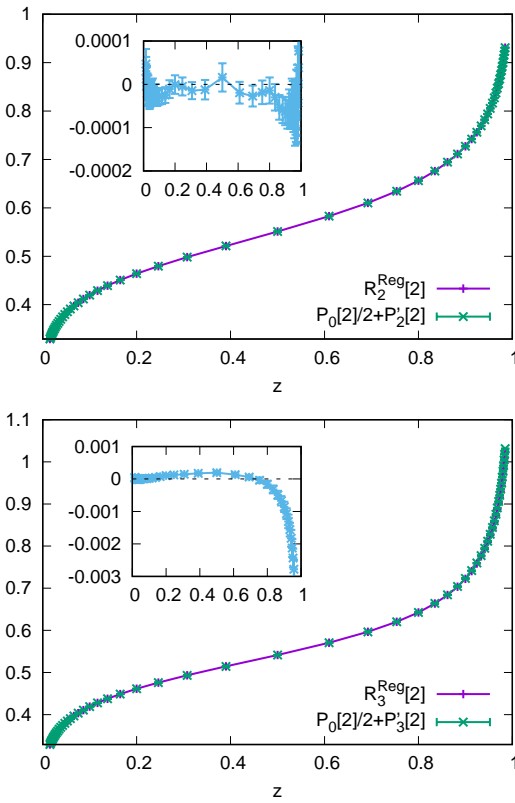

Figure 3.4: Regular parts of $R_2, R_3$ at $Q = 2$, compared with $\frac{1}{2}P_0[2] + P'_2[2]$ and $\frac{1}{2}P_0[2] + P'_3[2]$, as functions of $z$. The insets give the differences. The derivatives $P'_\sigma[2]$ are computed using the approximation $P'_\sigma[2] \sim \frac{P_\sigma[2.25] - P_\sigma[1.75]}{0.5}$.

The results from the fits agree with the following predictions:

| Prediction | Origin | Verification |
|---|---|---|
| $\beta_0^{(1)} = \beta_1^{(1)} = 2\Delta_{(0,\frac{1}{2})}$ | Eqs. (52), (27) | Table B.1 |
| $\beta_0^{(2)} \underset{Q \leq 2}{=} 2\Delta_{(2,0)}, \beta_0^{(2)} \underset{Q \geq 2}{=} 2\Delta_{(0,\frac{3}{2})}$ | Eq. (27), Fig. (2.1) | Table B.1 |
| $\beta_{2,3}^{(k \leq 4)} = \Delta_{(2,\frac{k-1}{2})} + \Delta_{(2,-\frac{k-1}{2})}$ | Eq. (27) | Tables B.3, B.4 |
| $\alpha_0^{(1)} = -\alpha_1^{(1)} = 2D_{(0,\frac{1}{2})}$ | Eqs. (43), (52) | Table B.1 |

(68)

In particular, we did not observe any contribution from the field $(1,0)$ in $P_1$, contrary to the first version of [12], but in agreement with their corrected statements.

Let us now discuss the asymptotic behaviour of connectivities in light of the relation (61) with the odd CFT. We first focus on the structure constants $D_{(2,\frac{1}{2})}, D_{(2,\frac{3}{2})}$, whose values are known analytically in the odd CFT. Let us compare these analytic values with fits for $P_2 - P_3$, since this combination is dominated by the states $(2, \frac{1}{2})$ and $(2, \frac{3}{2})$, as shown in Fig. 2.2. We introduce the fit

$$\frac{P_2(z) - P_3(z)}{Q - 2} \underset{z \in \mathbb{R}}{=} 4D_{(2,\frac{1}{2})}^{\text{fit}} \mathcal{F}_{\Delta_{(2,\frac{1}{2})}}^{(s)}(z) \mathcal{F}_{\Delta_{(2,-\frac{1}{2})}}^{(s)}(z) + 4D_{(2,\frac{3}{2})}^{\text{fit}} \mathcal{F}_{\Delta_{(2,\frac{3}{2})}}^{(s)}(z) \mathcal{F}_{\Delta_{(2,-\frac{3}{2})}}^{(s)}(z) \,, \tag{69}$$

and find that the parameters $D_{(2,\frac{1}{2})}^{\text{fit}}, D_{(2,\frac{3}{2})}^{\text{fit}}$ agree with the corresponding structure constants of the odd CFT. (See Table 3.6.) Again, the agreement worsens when $Q$ increases. We can

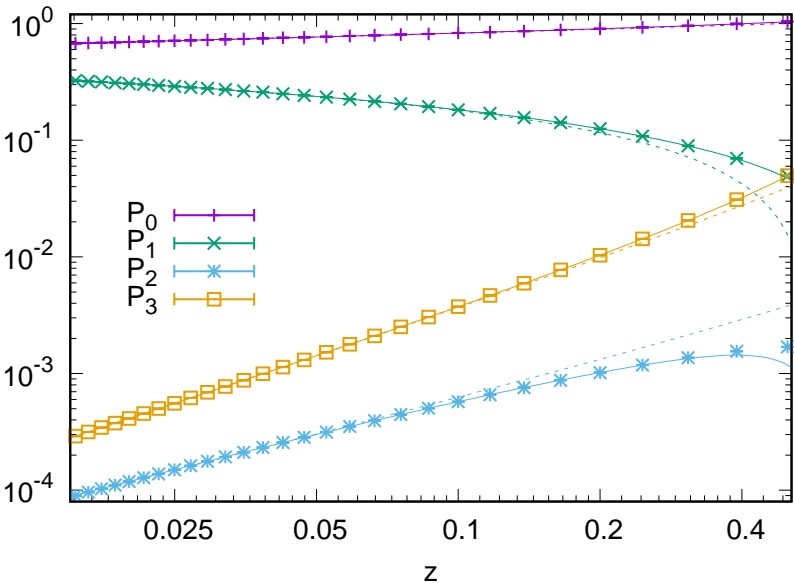

Figure 3.5: Monte-Carlo data for $P_\sigma(z)$ as a function of $z$ for $Q = 1$. We also plot the best fits of the type (67) with $K = 1$ term (dashed lines) and $K = 2$ terms (thin lines).

attribute this worsening to a loss of numerical precision, because we know that $D_{(2,\frac{1}{2})}$ and $D_{(2,\frac{3}{2})}$ are the exact structure constants in the case $Q = 3$, see Eq. (129).

Table 3.6: Comparison of the numerically determined parameters of the fit (69) with structure constants of the odd CFT.

| $Q$ | $D_{(2,\frac{1}{2})}$ | $D^{\text{fit}}_{(2,\frac{1}{2})}$ | $D_{(2,\frac{3}{2})}$ | $D^{\text{fit}}_{(2,\frac{3}{2})}$ |
|---|---|---|---|---|
| 1 | 0.02013 | 0.02011 | 0.00024 | 0.00025 |
| 1.25 | 0.02692 | 0.02685 | 0.00038 | 0.00041 |
| 1.38197 | 0.03247 | 0.03241 | 0.0005 | 0.00054 |
| 1.5 | 0.03979 | 0.03971 | 0.00065 | 0.00069(3) |
| 1.75 | 0.07736 | 0.07718 | 0.00146 | 0.00165(3) |
| 2.25 | −0.07074 | −0.0706 | −0.00171 | −0.00181(2) |
| 2.5 | −0.03329 | −0.03331 | −0.0009 | −0.00087(2) |
| 2.75 | −0.02065 | −0.02078 | −0.00062 | −0.00049(2) |
| 3.0 | −0.0142 | −0.01440 | −0.00047 | −0.00045 |
| 3.25 | −0.01023 | −0.01335(2) | −0.00038 | −0.00017(5) |

Therefore, the odd CFT's structure constants $D_{(0,\frac{1}{2})}$, $D_{(2,\frac{1}{2})}$ and $D_{(2,\frac{3}{2})}$ are numerically indistinguishable from Potts model's structure constants. It would however be implausible for the agreement to be exact, since the odd CFT cannot exactly describe connectivities. Nevertheless, we can use the relation with the odd CFT for making exact conjectures about other structure constants. To do this, we formulate the

> **Minimal completion hypothesis:** The spectrums of the combinations $R_\sigma$ (61) of Potts model connectivities is given by the corresponding odd CFT spectrums, plus whatever states are needed for cancelling the $Q$-poles, and nothing more.

This hypothesis relies on the idea that it is the $Q$-poles of the odd CFT four-point functions that

prevent the relation (61) from being exact, and that these poles can be cancelled by adding extra states from the full spectrum (27), as demonstrated in an example in [12]. However, not all states from the full spectrum are actually needed for cancelling the poles, and in particular the state $(2, 0)$ is never needed.

This diagonal primary state would indeed only be needed if its dimension coincided with the dimension of a pole of a conformal block from the odd CFT, which can be checked not to happen. For the state $(2, 0)$ to be absent from $R_2, R_3$ while being present in $P_0$ and $P_2, P_3$, we need the structure constants of that state to cancel in the combinations $\frac{1}{2}P_0 + \frac{1}{Q-2}P_{2,3}$. In the notations of Eq. (67), and in the regime $Q \in (0, 2)$ these cancellations read

$$\frac{1}{2}\alpha_0^{(2)} + \frac{1}{Q-2}\alpha_{2,3}^{(1)} = 0 \,. \tag{70}$$

For $Q > 2$, where $\Delta_{(2,0)} > \Delta_{(0,\frac{3}{2})}$, $\alpha_0^{(2)}$ is replaced by $\alpha_0^{(3)}$. We tested this prediction in Monte-Carlo data and found a reasonable agreement, although the tests become numerically more tricky as $Q$ increases, see Section B.2.

## 4  Special values of $Q$

We now focus on the integer values $Q = 0, 2, 3, 4$ of the number of states, which respectively correspond to the central charges $c = -2, \frac{1}{2}, \frac{4}{5}, 1$. On the one hand, we will review the known exact results for certain linear combinations of connectivities, which follow from their relations to spanning trees ($Q = 0$) or with spins correlation functions ($Q = 2, 3, 4$). On the other hand, we will show that these known exact results agree with four-point functions in the odd CFT in the cases $Q = 0, 3, 4$.

### 4.1  $Q = 0$: spanning trees

A graph $\mathcal{G}$ that contributes to the partition function (30) is characterized by the numbers of bonds, of clusters and of independent cycles. These quantities are not independent, as they satisfy the Euler relation:

$$\#\text{vertices} + \#\text{independent cycles} = \#\text{bonds} + \#\text{clusters} \,. \tag{71}$$

For a doubly periodic square lattice of size $L$, $\#\text{vertices} = \frac{1}{2}\#\text{edges} = L^2$. At criticality, the partition function can be written as

$$\mathcal{Z}^{\text{critical}} = \left(\frac{Q^{\frac{1}{4}}}{\sqrt{Q}+1}\right)^{\#\text{edges}} \sum_{\mathcal{G}} \left(\sqrt{Q}\right)^{\#\text{independent cycles}(\mathcal{G})+\#\text{clusters}(\mathcal{G})} \,. \tag{72}$$

Without loss of generality, we will neglect the $\mathcal{G}$-independent prefactor. In the limit $Q \to 0$, the leading contribution to the partition function is the number $\mathcal{Z}_{1-\text{forest}}$ of spanning trees, i.e. of configurations of one cluster with no cycles. The next contributions are the sum $\mathcal{Z}_{2-\text{forest}}$ over graphs made of two trees, and the sum $\mathcal{Z}_{1-\text{cycle}}$ over graphs made of one cluster with one cycle. The small $Q$ expansion therefore takes the form

$$\mathcal{Z}^{\text{critical}} = \mathcal{Z}_{1-\text{forest}} + \sqrt{Q}\left(\mathcal{Z}_{2-\text{forest}} + \mathcal{Z}_{1-\text{cycle}}\right) + O(Q) \,. \tag{73}$$

In particular, the scaling limits of our four-point connectivities behave as

$$P_\sigma = \delta_{\sigma,0} + O(\sqrt{Q}) \,, \quad (\sigma = 0, 1, 2, 3) \,. \tag{74}$$

Let us introduce the discrete Laplacian operator $A$ of a square lattice: this is a $L^2 \times L^2$ matrix with entries $A_{z_i, z_i} = 4$, $A_{z_i, z_j} = -1$ if $z_i$ and $z_j$ are nearest neighbours. Due to $\sum_j A_{z_i, z_j} = 0$, the matrix $A$ has a vanishing eigenvalue. According to the matrix-tree theorem,

$$\mathcal{Z}_{1-\mathrm{forest}} = \det{}'[A] = \det[A(z_i; z_i)] \,, \tag{75}$$

where $\det'[A]$ is the product of the non-vanishing eigenvalues of $A$, and $\det[A(z_i; z_i)]$ is the determinant of the $A$ after removing $z_i$'s row and $z_i$'s column. Given the pairs of points $(z_1, z_2)$ and $(z_3, z_4)$, let $\mathcal{Z}_{12,34}$ be the sum over graphs made of two disconnected trees, one containing $z_1$ and $z_2$ and the other $z_3$ and $z_4$. The extension of the matrix-tree theorem reads

$$\mathcal{Z}_{13,24} - \mathcal{Z}_{14,23} = \det\left[A(z_1, z_2; z_3, z_4)\right] \,, \tag{76}$$

where $\det\left[A(z_1, z_2; z_3, z_4)\right]$ is the determinant of $A$ after removing $z_1$ and $z_2$'s rows, and $z_3$ and $z_4$'s columns. We therefore obtain

$$p_{13,24} - p_{14,23} = \sqrt{Q}\frac{\mathcal{Z}_{13,24} - \mathcal{Z}_{14,23}}{\mathcal{Z}_{1-\mathrm{forest}}} + O(Q) = \sqrt{Q}\frac{\det\left[A(z_1, z_2; z_3, z_4)\right]}{\det'[A]} + O(Q)\,. \tag{77}$$

The scaling limit $P_2 - P_3$ of this expression is computed in Appendix C.1, with the result

$$P_2 - P_3 = \frac{1}{4\pi}\sqrt{Q}\left(\log(1-z) + \log(1-\bar{z})\right) + O(Q)\,, \tag{78}$$

which reproduces a result from Appendix B.4 of [12], while determining the numerical prefactor. From Eq. (74), we also deduce

$$P_0 = 1 + O(\sqrt{Q}) \quad, \quad P_1, P_2, P_3 = O(\sqrt{Q})\,. \tag{79}$$

## 4.2 $Q = 0$: limit of the odd CFT

Let us determine the behaviour of the structure constants $D_{(r,s)}$ (19) in the limit $Q \to 0$ i.e. $\beta^2 \to \frac{1}{2}$. This limit is similar to the limits where the odd CFT reduces to a minimal model [11], except that there is no minimal model at $c = -2$, as the Kac table is empty. Nevertheless, as in the case of minimal models, the structure constants have zeros of increasing orders as $r, s$ increase. To see this, we use the relation $\frac{1}{\beta} = 2\beta$ to write the arguments of the special functions as multiples of $\beta$. Then $\Gamma_\beta(-n\beta)$ has a pole of order $1 + \lfloor\frac{n}{2}\rfloor$ for $n \in \mathbb{N}$, while $\Upsilon_\beta(-n\beta)$ has a zero of order $1 + \lfloor\frac{n}{2}\rfloor$, and $\Upsilon_\beta(x) = \Upsilon_\beta(3\beta - x)$. It follows that $D_{(r,s)}$ has a first-order pole from the prefactor $\frac{1}{\Gamma(2-\frac{1}{\beta^2})\Upsilon_\beta(\frac{3}{2\beta})^2}$, together with $(r,s)$-dependent poles and zeros. Overall, $D_{(0,\frac{1}{2})}$ has a finite limit, $D_{(2,\pm\frac{1}{2})}$ has a first-order zero, and all the other structure constants have zeros of higher orders. In terms of the variable $Q$, the zero of $D_{(2,\pm\frac{1}{2})}$ has the order $\frac{1}{2}$, due to the relation

$$\beta^2 - \frac{1}{2} \underset{Q\to 0}{=} \frac{1}{2\pi}\sqrt{Q} + O(Q^{\frac{5}{2}})\,. \tag{80}$$

After tedious but straightforward calculations, we find

$$D_{(0,\frac{1}{2})} \underset{Q\to 0}{=} \frac{1}{2} + O(\sqrt{Q})\,, \tag{81}$$

$$D_{(2,\frac{1}{2})} \underset{Q\to 0}{=} \frac{1}{16\pi}\sqrt{Q} + O(Q)\,. \tag{82}$$

Let us determine the conformal blocks $\mathcal{F}^{(s)}_{\Delta}(z)$ for $\Delta \in \left\{\Delta_{(0,\frac{1}{2})}, \Delta_{(2,\frac{1}{2})}, \Delta_{(2,-\frac{1}{2})}\right\}$. Since $\Delta_{(0,\frac{1}{2})} = \Delta_{(2,\frac{1}{2})} = \Delta_{(1,1)} = 0$, the first two cases are trivial blocks,

$$\mathcal{F}^{(s)}_{\Delta_{(0,\frac{1}{2})}}(z) = \mathcal{F}^{(s)}_{\Delta_{(2,\frac{1}{2})}}(z) = \begin{array}{c} (1,1) \\[-4pt] \diagdown \end{array} \underset{(1,1)}{\underbrace{\qquad}} \begin{array}{c} (1,1) \\[-4pt] \diagup \end{array} = 1 \, . \tag{83}$$

On the other hand, since $\Delta_{(2,-\frac{1}{2})} = \Delta_{(1,-1)} = \Delta_{(5,1)} = 1$, the block $\mathcal{F}^{(s)}_{\Delta_{(2,-\frac{1}{2})}}(z)$ cannot appear in the four-point function of the identity field $\left\langle V_{(1,1)} V_{(1,1)} V_{(1,1)} V_{(1,1)} \right\rangle$. However, we also have $\Delta_{(0,\frac{1}{2})} = \Delta_{(3,1)}$, and our block can be written as

$$\mathcal{F}^{(s)}_{\Delta_{(2,-\frac{1}{2})}}(z) = \begin{array}{c} (3,1) \\[-4pt] \diagdown \end{array} \underset{(5,1)}{\underbrace{\qquad}} \begin{array}{c} (3,1) \\[-4pt] \diagup \end{array} \tag{84}$$

We will determine our block by solving a third-order BPZ differential equation. Rather than a direct derivation from the singular vector, we will use knowledge of the fusion rules for deriving this equation, in the spirit of [26]. The fusion rules determine the characteristic exponents of our Fuchsian differential equation at each one of the three singularities $z = 0, 1, \infty$. In principle this is not enough for completely determining the equation, as our third-order system is not rigid. In our case, requiring that $\mathcal{F}^{(s)}_{\Delta_{(0,\frac{1}{2})}}(z) = 1$ is also a solution provides the missing constraint. Using this constraint, we write the equation as

$$\left\{ \frac{\partial^2}{\partial z^2} + \left( \frac{\lambda}{z} + \frac{\mu}{z-1} \right) \frac{\partial}{\partial z} + \frac{\nu}{z^2} + \frac{\rho}{z(z-1)} + \frac{\sigma}{(z-1)^2} \right\} \frac{\partial}{\partial z} \mathcal{F}^{(s)}(z) = 0 \, , \tag{85}$$

for some coefficients $\lambda, \mu, \nu, \rho, \sigma$. From the fusion rules, we know that the characteristic exponents at $z = 0, 1, \infty$ are $\{\alpha\} = \{0, 0, 1\}$. Inserting the ansatz $\mathcal{F}^{(s)}(z) \underset{z \to 0}{=} z^{\alpha}(1 + O(z))$, we find that the characteristic exponents at $z = 0$ are solutions of

$$\alpha\left((\alpha - 1)(\alpha - 2) + \lambda(\alpha - 1) + \nu\right) = 0 \, . \tag{86}$$

In order for this polynomial to have the required roots, we need $\lambda = 2$ and $\nu = 0$. Similarly, in order to have the right exponents at $z = 1$, we need $\mu = 2$ and $\sigma = 0$. Finally, inserting $\mathcal{F}^{(s)}(z) \underset{z \to \infty}{=} z^{-\alpha}(1 + O(z^{-1}))$ near $z = \infty$, we find the equation

$$\alpha\left((\alpha + 1)(\alpha + 2) - (\lambda + \mu)(\alpha + 1) + \nu + \rho + \sigma\right) = 0 \, , \tag{87}$$

and this determines $\rho = 2$. Summarizing, our third-order BPZ equation is

$$\left\{ \frac{1}{2} \frac{\partial^2}{\partial z^2} + \left( \frac{1}{z} + \frac{1}{z-1} \right) \frac{\partial}{\partial z} + \frac{1}{z(z-1)} \right\} \frac{\partial}{\partial z} \mathcal{F}^{(s)}(z) = 0 \, , \tag{88}$$

and its three independent solutions are $\mathcal{F}^{(s)}(z) = 1, \log z, \log(1-z)$. The only solution that behaves as $\mathcal{F}^{(s)}(z) \underset{z \to 0}{=} z(1 + O(z))$ is

$$\mathcal{F}^{(s)}_{\Delta_{(2,-\frac{1}{2})}}(z) = \log(1-z) \, . \tag{89}$$

Knowing the behaviour of the structure constants and conformal blocks as $Q \to 0$, we now deduce the behaviour of the four-point functions $R_1, R_2, R_3$ (16)-(18). We first eliminate all the terms that behave as $O(Q)$:

$$R_2 \underset{Q \to 0}{=} D_{(0,\frac{1}{2})} \mathcal{F}^{(s)}_{\Delta_{(0,\frac{1}{2})}}(z) \mathcal{F}^{(s)}_{\Delta_{(0,\frac{1}{2})}}(\bar{z})$$
$$+ D_{(2,\frac{1}{2})} \left( \mathcal{F}^{(s)}_{\Delta_{(2,-\frac{1}{2})}}(z) \mathcal{F}^{(s)}_{\Delta_{(2,\frac{1}{2})}}(\bar{z}) + \mathcal{F}^{(s)}_{\Delta_{(2,\frac{1}{2})}}(z) \mathcal{F}^{(s)}_{\Delta_{(2,-\frac{1}{2})}}(\bar{z}) \right) + O(Q) . \quad (90)$$

More explicitly, this is

$$R_2 \underset{Q \to 0}{=} \frac{1}{2} + \frac{1}{16\pi} \sqrt{Q} \Big( \log(1-z) + \log(1-\bar{z}) \Big) + O(\sqrt{Q}) . \quad (91)$$

We are now neglecting $O(\sqrt{Q})$ terms from the expansions of $D_{(0,\frac{1}{2})}$ and of the block $\mathcal{F}^{(s)}_{\Delta_{(0,\frac{1}{2})}}(z)$. Such terms are not easy to compute, and it is clear that the block has a logarithmic singularity at $z = 0$ at this order. Neglecting these terms however does not make our $O(\sqrt{Q})$ terms from the $(2, \frac{1}{2})$ channel meaningless, because it is the $(2, \frac{1}{2})$ channel that dominates in differences of correlation functions, for example

$$R_2 - R_3 \underset{Q \to 0}{=} \frac{1}{8\pi} \sqrt{Q} \Big( \log(1-z) + \log(1-\bar{z}) \Big) + O(Q) . \quad (92)$$

Comparing with equations (78) and (79) for the behaviour of connectivities, we find that the relation (61) between connectivities and correlation functions exactly holds at $Q = 0$.

### 4.3 $Q = 2, 3, 4$: spin correlation functions

For $Q = 2, 3, 4$, the Potts model can be defined in terms of spins $s(x)$, sitting at each lattice site $x$ and taking $Q$ values, $s(x) = 1, \ldots, Q$. The partition function is

$$\mathcal{Z} = \prod_{\langle x, y \rangle} \exp \left[ -J \delta_{s(x), s(y)} \right] , \quad (93)$$

where the product is over pairs of neighbouring sites. The model is invariant under permutations of the $Q$ values of the spins. It has a critical point at

$$J_c(Q) = \log \left( 1 + \sqrt{Q} \right) , \quad (94)$$

which separates a ferromagnetic phase from a paramagnetic phase. We introduce the spin observables

$$\sigma_\alpha(x) = Q \delta_{s(x), \alpha} - 1 \quad \text{where} \quad (\alpha = 1, \ldots, Q) , \quad (95)$$

which obey the relation $\sum_{\alpha=1}^{Q} \sigma_\alpha(x) = 0$. This relation, and permutation symmetry, lead to linear relations between correlators of spin observables. Modulo such linear relations, all two- and three-point functions are proportional to two- and three-point functions of the type $\langle \sigma_\alpha \sigma_\alpha \rangle$ and $\langle \sigma_\alpha \sigma_\alpha \sigma_\alpha \rangle$, where the latter actually vanishes in the case $Q = 2$. Furthermore, correlators of spin observables can be expressed in terms of cluster connectivities. In the case of two- and three-point functions, the relations are

$$\langle \sigma_\alpha \sigma_\alpha \rangle = (Q-1) p_{12} , \quad (96)$$
$$\langle \sigma_\alpha \sigma_\alpha \sigma_\alpha \rangle = (Q-1)(Q-2) p_{123} . \quad (97)$$

All four-point functions are linear combinations of the four basic functions $\langle\sigma_\alpha\sigma_\alpha\sigma_\alpha\sigma_\alpha\rangle$, $\langle\sigma_\beta\sigma_\beta\sigma_\alpha\sigma_\alpha\rangle$, $\langle\sigma_\beta\sigma_\alpha\sigma_\beta\sigma_\alpha\rangle$, $\langle\sigma_\beta\sigma_\alpha\sigma_\alpha\sigma_\beta\rangle$ with $\beta\neq\alpha$. The expressions of these basic four-point functions as linear combinations of the cluster connectivities can be found in [7]. Let us write these expressions in the cases $Q=2,3,4$.

For $Q=2$, our four basic four-point functions are all equal since $\sigma_2=-\sigma_1$, and the relation with connectivities is

$$\langle\sigma_1\sigma_1\sigma_1\sigma_1\rangle = p_{1234} + p_{12,34} + p_{13,24} + p_{14,23} \,. \tag{98}$$

For $Q=3$, it is convenient to use the basis of operators $\{\omega_1,\omega_2\}$ such that

$$\sigma_1 = e^{\frac{2\pi i}{3}}\omega_1 + e^{\frac{4\pi i}{3}}\omega_2 \quad, \quad \sigma_2 = e^{\frac{4\pi i}{3}}\omega_1 + e^{\frac{2\pi i}{3}}\omega_2 \quad, \quad \sigma_3 = \omega_1 + \omega_2 \,. \tag{99}$$

In terms of these operators, permutations of $\sigma_\alpha$ are generated by the permutation $\omega_1 \leftrightarrow \omega_2$, and the $\mathbb{Z}_3$ action

$$\omega_\alpha \to e^{\frac{2\pi i}{3}\alpha}\omega_\alpha \,. \tag{100}$$

At the level of four-point functions, the change of bases reads

$$\langle\sigma_\alpha\sigma_\alpha\sigma_\alpha\sigma_\alpha\rangle = 2\langle\omega_1\omega_1\omega_2\omega_2\rangle + 2\langle\omega_1\omega_2\omega_1\omega_2\rangle + 2\langle\omega_1\omega_2\omega_2\omega_1\rangle \,, \tag{101}$$

$$\langle\sigma_\beta\sigma_\beta\sigma_\alpha\sigma_\alpha\rangle = -\langle\omega_1\omega_1\omega_2\omega_2\rangle + 2\langle\omega_1\omega_2\omega_1\omega_2\rangle + 2\langle\omega_1\omega_2\omega_2\omega_1\rangle \,. \tag{102}$$

Due to the $\mathbb{Z}_3$ symmetry and the symmetry under $\omega_1 \leftrightarrow \omega_2$, the three correlators that appear here form a basis of four-point correlators of $\omega_1,\omega_2$. Their relations with connectivities are

$$\langle\omega_1\omega_1\omega_2\omega_2\rangle = p_{1234} + p_{13,24} + p_{14,23} \,, \tag{103}$$

$$\langle\omega_1\omega_2\omega_1\omega_2\rangle = p_{1234} + p_{12,34} + p_{14,23} \,, \tag{104}$$

$$\langle\omega_1\omega_2\omega_2\omega_1\rangle = p_{1234} + p_{12,34} + p_{13,24} \,. \tag{105}$$

For $Q=4$, it is convenient to introduce yet another basis of operators: the spins $\tau_1$ and $\tau_2$ of two Ising models, which are coupled in the Ashkin–Teller model, which is itself equivalent to the $Q=4$ Potts model. In terms of these spins, the partition function reads [27]

$$\mathcal{Z} = \sum_{\langle x,y\rangle} e^{K_2[\tau_1(x)\tau_1(y)+\tau_2(x)\tau_2(y)]+K_4[\tau_1(x)\tau_1(y)\tau_2(x)\tau_2(y)]} \,. \tag{106}$$

This model is critical for $e^{-2K_4} = \sinh 2K_2$, with the $Q=4$ Potts model corresponding to the case $K_2 = K_4 = \frac{J_c(4)}{4} = \frac{\log 3}{4}$, where the Potts model's critical coupling $J_c(Q)$ was given in Eq. (94). The relation between the Ising spins and the operators $\sigma_\alpha$ is

$$4\tau_1 = \sigma_1 - \sigma_2 - \sigma_3 + \sigma_4 \,, \tag{107}$$

$$4\tau_2 = \sigma_1 + \sigma_2 - \sigma_3 - \sigma_4 \,, \tag{108}$$

$$4\tau_1\tau_2 = -\sigma_1 + \sigma_2 - \sigma_3 + \sigma_4 \,. \tag{109}$$

A basis of four linearly independent four-point functions, and their relations with connectivities, are

$$\langle\tau_1\tau_1\tau_1\tau_1\rangle = p_{1234} + p_{12,34} + p_{13,24} + p_{14,23} \,, \tag{110}$$

$$\langle\tau_1\tau_1\tau_2\tau_2\rangle = p_{1234} + p_{12,34} \,, \tag{111}$$

$$\langle\tau_1\tau_2\tau_1\tau_2\rangle = p_{1234} + p_{13,24} \,, \tag{112}$$

$$\langle\tau_1\tau_2\tau_2\tau_1\rangle = p_{1234} + p_{14,23} \,. \tag{113}$$

In the remainder of this section, we will exactly compute the spin correlation functions for $Q=2,3,4$, and deduce the corresponding combinations of connectivities: one combination for $Q=2$, three combinations for $Q=3$, and all four connectivities for $Q=4$. In the cases $Q=3,4$, we will compare these combinations with correlation functions in the odd CFT.

### 4.4 $Q = 2$: minimal model

In the critical limit, the spin four-point function $\langle \sigma_1 \sigma_1 \sigma_1 \sigma_1 \rangle$ (98) is described by the Ising minimal model, whose central charge is $c = \frac{1}{2}$. In particular, the spin field is

$$\sigma_1 = V^D_{(0,\frac{1}{2})} = V^D_{(2,1)}, \qquad \text{with} \qquad \Delta_{(0,\frac{1}{2})} = \frac{1}{16}. \tag{114}$$

Actually, in the case $Q = 2$, we know the exact expression of $\langle \sigma_1 \sigma_1 \sigma_1 \sigma_1 \rangle$ not only on the plane, but also on the torus. Since we use the torus geometry as the large distance regulator in our Monte-Carlo calculations, this provides us with exact predictions not only for the large distance limit, but also for the corrections thereto. These predictions take the form [28, 29]

$$\left\langle \sigma_1(ir)\sigma_1(0)\sigma_1(\lambda r)\sigma_1(i\lambda + ir) \right\rangle_L = \frac{c_0(z)}{\sqrt{r}} \left[ 1 + c_1(z)\frac{r}{L} + c_4(z)\left(\frac{r}{L}\right)^4 + \cdots \right], \tag{115}$$

where $L$ is the size of the torus, and $z$ is the cross-ratio (45). This is a special case of the finite size corrections (58). The leading factor $c_0(z)$ is the sphere four-point function

$$c_0(z) = \sum_{\sigma=0}^{3} P_\sigma(z) = z^{\frac{1}{4}} \left\langle \sigma_1(0)\sigma_1(z)\sigma_1(1)\sigma_1(\infty) \right\rangle, \tag{116}$$

whose explicit expression is [30]

$$\left\langle \sigma_1(0)\sigma_1(z)\sigma_1(1)\sigma_1(\infty) \right\rangle = \mathcal{F}^{(s)}_{(1,1)}(z)\mathcal{F}^{(s)}_{(1,1)}(\bar{z}) + \frac{1}{4}\mathcal{F}^{(s)}_{(3,1)}(z)\mathcal{F}^{(s)}_{(3,1)}(\bar{z}),$$

$$= |z|^{-\frac{1}{4}}|1-z|^{-\frac{1}{4}} \left( \frac{1}{2}\left|\sqrt{1+\sqrt{1-z}}\right|^2 + \frac{1}{4}\left|\sqrt{1+\sqrt{z}} - \sqrt{1-\sqrt{z}}\right|^2 \right). \tag{117}$$

This expression simplifies considerably for $z \in (0, 1)$,

$$c_0(z) \underset{z\in(0,1)}{=} (1-z)^{-\frac{1}{4}}. \tag{118}$$

Moreover, the coefficients of the first two subleading terms have the form

$$c_1(z) = 0.97772562(1-z)^{\frac{1}{2}}, \quad c_4(z) = 6.82586\frac{1-z}{z}. \tag{119}$$

Notice that in the sum $P_0 + P_1 + P_2 + P_3$ the leading $s$-channel is the identity $\Delta_{(1,1)} = 0$ and the sub-leading is the energy field $\Delta_{(3,1)} = \frac{1}{2}$.

### 4.5 $Q = 3$: minimal model

In the case $Q = 3$, let us compute the spin correlation functions by using the $\mathcal{W}_3$ minimal model. While less elementary than the Virasoro algebra, the W-algebra $\mathcal{W}_3$ has a manifest $\mathbb{Z}_3$ symmetry, which matches the $\mathbb{Z}_3$ symmetry (100) of the $Q = 3$ Potts model. In particular, the fields $\omega_1, \omega_2$ whose four-point functions we want to compute, correspond to $\mathcal{W}_3$ primary fields of the types $V_{(21)(11)}, V_{(12)(11)}$, with the same conformal dimension $\Delta_{(0,\frac{1}{2})} = \frac{1}{15}$ but with opposite $\mathbb{Z}_3$ charges. (See Appendix C.2 for a reminder on the relevant $\mathcal{W}_3$ minimal model, and [31] for an earlier appearance of this correspondence.) This leads to relatively simple expressions for our four-point functions, in particular

$$\left\langle \omega_1(0)\omega_2(z)\omega_2(1)\omega_1(\infty) \right\rangle = |z|^{-4\Delta_{(0,\frac{1}{2})}}|1-z|^{-2\Delta_{(0,\frac{1}{2})}}$$

$$\times \left( \left| {}_2F_1\left(-\frac{1}{5}, \frac{1}{5}; \frac{3}{5}; z\right) \right|^2 + d_{(2,\frac{1}{2})}|z|^{12\Delta_{(0,\frac{1}{2})}} \left| {}_2F_1\left(\frac{1}{5}, \frac{3}{5}; \frac{7}{5}; z\right) \right|^2 \right), \tag{120}$$

where the structure constant $d_{(2,\frac{1}{2})}$ is determined by the single-valuedness of our four-point function. This structure constant can be written in terms of the structure constant $D_{(0,\frac{1}{2})}$ of the odd CFT, whose expression we will shortly give, as

$$d_{(2,\frac{1}{2})} = \frac{1}{2} D_{(0,\frac{1}{2})} \,. \tag{121}$$

Let us compare this with the odd CFT, which for $Q = 3$ i.e. $c = \frac{4}{5}$ reduces to a D-series minimal model. The minimal model in question has the following non-diagonal primary fields and four-point structure constants (19):

| Fields | Left and right dimensions | Structure constants |
|---|---|---|
| $V^N_{(0,\frac{1}{2})}$ | $\left(\frac{1}{15}, \frac{1}{15}\right)$ | $D_{(0,\frac{1}{2})} = \frac{1}{2} \frac{\Gamma[\frac{1}{5}]\Gamma[\frac{3}{5}]^3}{\Gamma[\frac{4}{5}]\Gamma[\frac{2}{5}]^3}$ |
| $V^N_{(0,\frac{3}{2})}$ | $\left(\frac{2}{3}, \frac{2}{3}\right)$ | $D_{(0,\frac{3}{2})} = \frac{1}{18}$ |
| $V^N_{(2,\pm\frac{1}{2})}$ | $\left(\frac{2}{5}, \frac{7}{5}\right)$ | $D_{(2,\frac{1}{2})} = \frac{1}{42} D_{(0,\frac{1}{2})}$ |
| $V^N_{(2,\pm\frac{3}{2})}$ | $(0, 3)$ | $D_{(2,\frac{3}{2})} = \frac{1}{2106}$ |

$$\tag{122}$$

The two fields $V^{D,N}_{(0,\frac{1}{2})}$ must be linear combinations of $\omega_1, \omega_2$, which span the space of Virasoro-primary fields with conformal dimensions $\left(\frac{1}{15}, \frac{1}{15}\right)$. In order to determine the coefficients of the linear relation, we normalize the fields by assuming the two-point functions

$$\begin{cases} \langle \omega_1 \omega_2 \rangle = 1 \,, \\ \langle \omega_1 \omega_1 \rangle = 0 \,, \\ \langle \omega_2 \omega_2 \rangle = 0 \,, \end{cases} \qquad \begin{cases} \langle V^D V^D \rangle = 1 \,, \\ \langle V^N V^N \rangle = 1 \,, \\ \langle V^D V^N \rangle = 0 \,. \end{cases} \tag{123}$$

This determines the linear relation to be

$$V^D = \frac{1}{\sqrt{2}} (\omega_1 + \omega_2) \qquad , \qquad V^N = \frac{i}{\sqrt{2}} (\omega_1 - \omega_2) \,, \tag{124}$$

up to the rescalings $\begin{cases} \omega_1 \to \lambda \omega_1 \\ \omega_2 \to \lambda^{-1} \omega_2 \end{cases}$ , which leave two-point functions invariant. This residual ambiguity is eliminated by the symmetry $\omega_1 \leftrightarrow \omega_2$, which has to correspond to the symmetry $V^N \to -V^N$, in other words to the conservation of diagonality of Section 2.1. In particular, at the level of three-point functions, we have

$$\langle \omega_1 \omega_1 \omega_1 \rangle = \langle \omega_2 \omega_2 \omega_2 \rangle = \sqrt{2} \langle V^D V^D V^D \rangle. \tag{125}$$

From the above equation and using (97) with $\langle \sigma_0 \sigma_0 \sigma_0 \rangle = 2 \langle \omega_1 \omega_1 \omega_1 \rangle$, one can prove that Eq. (43) is exact for $Q = 3$. When it comes to four-point functions, we find

$$2 \langle V^D V^D V^N V^N \rangle = \langle \omega_1 \omega_2 \omega_1 \omega_2 \rangle + \langle \omega_1 \omega_2 \omega_2 \omega_1 \rangle - \langle \omega_1 \omega_1 \omega_2 \omega_2 \rangle \,, \tag{126}$$

$$2 \langle V^D V^D V^D V^D \rangle = \langle \omega_1 \omega_2 \omega_1 \omega_2 \rangle + \langle \omega_1 \omega_2 \omega_2 \omega_1 \rangle + \langle \omega_1 \omega_1 \omega_2 \omega_2 \rangle \,. \tag{127}$$

Therefore, the expressions (103)-(105) for the correlation functions of $\omega_1, \omega_2$ in terms of connectivities lead to the following expressions for the correlation functions $R_\sigma$ (13)-(15),

$$R_\sigma(z) = \tfrac{1}{2} P_0(z) + P_\sigma(z) \,. \tag{128}$$

This shows that the relation (61), which we argued was only approximate, becomes exact for $Q = 3$. The correlators $R_\sigma(z)$ are given by the expressions (16)-(18), where the nominally infinite sums truncate to finitely many terms. In particular,

$$P_2(z) - P_3(z) = 2D_{(2,\frac{1}{2})} \sum_{\pm} \mathcal{F}^{(s)}_{\Delta_{(2,\pm\frac{1}{2})}}(z) \bar{\mathcal{F}}^{(s)}_{\Delta_{(2,\mp\frac{1}{2})}}(\bar{z})$$
$$+ 2D_{(2,\frac{3}{2})} \sum_{\pm} \mathcal{F}^{(s)}_{\Delta_{(2,\pm\frac{3}{2})}}(z) \bar{\mathcal{F}}^{(s)}_{\Delta_{(2,\mp\frac{3}{2})}}(\bar{z}) . \quad (129)$$

Finally, let us give some more details on the relation between the $\mathcal{W}_3$ and D-series minimal models. We want to check that in the relation

$$\langle \omega_1 \omega_2 \omega_2 \omega_1 \rangle = \langle V^D V^D V^D V^D \rangle - \langle V^D V^N V^N V^D \rangle , \quad (130)$$

we can recover the formula (120) for the left-hand side, from the expansion of the right-hand side in terms of $s$-channel conformal blocks. The expansion of $\langle V^D V^N V^N V^D \rangle$ is given in Eq. (18), where only six terms are non-vanishing,

$$\langle V^D V^N V^N V^D \rangle = D_{(0,\frac{1}{2})} \mathcal{F}^{(s)}_{\Delta_{(0,\frac{1}{2})}} \bar{\mathcal{F}}^{(s)}_{\Delta_{(0,\frac{1}{2})}} + D_{(0,\frac{3}{2})} \mathcal{F}^{(s)}_{\Delta_{(0,\frac{3}{2})}} \bar{\mathcal{F}}^{(s)}_{\Delta_{(0,\frac{3}{2})}}$$
$$- D_{(2,\frac{1}{2})} \left( \mathcal{F}^{(s)}_{\Delta_{(2,\frac{1}{2})}} \bar{\mathcal{F}}^{(s)}_{\Delta_{(2,-\frac{1}{2})}} + \mathcal{F}^{(s)}_{\Delta_{(2,-\frac{1}{2})}} \bar{\mathcal{F}}^{(s)}_{\Delta_{(2,\frac{1}{2})}} \right)$$
$$- D_{(2,\frac{3}{2})} \left( \mathcal{F}^{(s)}_{\Delta_{(2,\frac{3}{2})}} \bar{\mathcal{F}}^{(s)}_{\Delta_{(2,-\frac{3}{2})}} + \mathcal{F}^{(s)}_{\Delta_{(2,-\frac{3}{2})}} \bar{\mathcal{F}}^{(s)}_{\Delta_{(2,\frac{3}{2})}} \right) . \quad (131)$$

The expansion of $\langle V^D V^D V^D V^D \rangle$ involves the corresponding six diagonal primary fields. As a consequence of general relations between diagonal and non-diagonal CFTs [13], the product of the structure constants of $V^D_{(r,s)}$ and $V^D_{(r,-s)}$ must be the square of the structure constant of $V^N_{(r,s)}$ in $\langle V^D V^N V^N V^D \rangle$, i.e. $D^2_{(r,s)}$. Moreover, the structure constant of the identity field $V^D_{(2,\frac{3}{2})}$ is one. This leads to

$$\langle V^D V^D V^D V^D \rangle = D_{(0,\frac{1}{2})} \mathcal{F}^{(s)}_{\Delta_{(0,\frac{1}{2})}} \bar{\mathcal{F}}^{(s)}_{\Delta_{(0,\frac{1}{2})}} + D_{(0,\frac{3}{2})} \mathcal{F}^{(s)}_{\Delta_{(0,\frac{3}{2})}} \bar{\mathcal{F}}^{(s)}_{\Delta_{(0,\frac{3}{2})}}$$
$$+ d_{(2,\frac{1}{2})} \mathcal{F}^{(s)}_{\Delta_{(2,\frac{1}{2})}} \bar{\mathcal{F}}^{(s)}_{\Delta_{(2,\frac{1}{2})}} + \frac{D^2_{(2,\frac{1}{2})}}{d_{(2,\frac{1}{2})}} \mathcal{F}^{(s)}_{\Delta_{(2,-\frac{1}{2})}} \bar{\mathcal{F}}^{(s)}_{\Delta_{(2,-\frac{1}{2})}}$$
$$+ \mathcal{F}^{(s)}_{\Delta_{(2,\frac{3}{2})}} \bar{\mathcal{F}}^{(s)}_{\Delta_{(2,\frac{3}{2})}} + D^2_{(2,\frac{3}{2})} \mathcal{F}^{(s)}_{\Delta_{(2,-\frac{3}{2})}} \bar{\mathcal{F}}^{(s)}_{\Delta_{(2,-\frac{3}{2})}} , \quad (132)$$

where we introduced the coefficient $d_{(2,\frac{1}{2})}$. Combining the previous two equations, we obtain

$$\langle \omega_1 \omega_2 \omega_2 \omega_1 \rangle = d_{(2,\frac{1}{2})} \left| \mathcal{F}^{(s)}_{\Delta_{(2,\frac{1}{2})}} - \frac{D_{(2,\frac{1}{2})}}{d_{(2,\frac{1}{2})}} \mathcal{F}^{(s)}_{\Delta_{(2,-\frac{1}{2})}} \right|^2 + \left| \mathcal{F}^{(s)}_{\Delta_{(2,\frac{3}{2})}} - D_{(2,\frac{3}{2})} \mathcal{F}^{(s)}_{\Delta_{(2,-\frac{3}{2})}} \right|^2 . \quad (133)$$

We find that this agrees with Eq. (120), thanks to the following identities between $\mathcal{W}_3$ and Virasoro conformal blocks:

$$(1-z)^{-\frac{1}{15}} z^{\frac{4}{15}} {}_2F_1\left(\frac{1}{5}, \frac{3}{5}; \frac{7}{5}; z\right) = \mathcal{F}^{(s)}_{\Delta_{(2,\frac{1}{2})}} - \frac{1}{21} \mathcal{F}^{(s)}_{\Delta_{(2,-\frac{1}{2})}} , \quad (134)$$

$$(1-z)^{-\frac{1}{15}} z^{-\frac{2}{15}} {}_2F_1\left(-\frac{1}{5}, \frac{1}{5}; \frac{3}{5}; z\right) = \mathcal{F}^{(s)}_{\Delta_{(2,\frac{3}{2})}} - \frac{1}{2106} \mathcal{F}^{(s)}_{\Delta_{(2,-\frac{3}{2})}} , \quad (135)$$

together with the formulas (122) for the structure constants, plus the relation (121).

## 4.6 $Q = 4$: Ashkin–Teller model

In the case $Q = 4$, the four-point functions of the fields $\tau_1, \tau_2$ (107)-(108) we computed by Al. Zamolodchikov [32]. In order to compare them with the odd CFT, we specialize the structure constants (19) to the case $\beta = 1$, and find

$$D_{(r,s)} \underset{c=1}{=} 16^{-\frac{1}{2}(r^2+s^2)}, \quad \text{in particular } D_{(0,\frac{1}{2})} = \frac{1}{\sqrt{2}}. \tag{136}$$

(Up to the $r, s$-independent prefactor, this formula comes from [13].) Furthermore, the relevant conformal blocks take the form [32]

$$\mathcal{F}_\Delta^{(s)}(z) \underset{c=1}{=} z^{-\frac{1}{8}}(1-z)^{-\frac{1}{8}} \frac{(16q(z))^\Delta}{\theta_3(q(z))}, \tag{137}$$

where $q(z) = \exp{-\pi \frac{F(\frac{1}{2},\frac{1}{2},1,1-z)}{F(\frac{1}{2},\frac{1}{2},1,z)}}$ is the nome and $\theta_3(q) = \sum_{n\in\mathbb{Z}} q^{n^2}$ is a Jacobi theta function. We obtain relations of the type

$$R_1 = \frac{1}{2}\langle \tau_1 \tau_1 \tau_2 \tau_2 \rangle. \tag{138}$$

Given Eq. (111), this agrees with our relation (61) between correlation functions $R_\sigma$ in the odd CFT, and scaling limits of connectivities. For $Q = 4$, this relation is therefore exact. Moreover, according to Eq. (110)-(113), we have

$$p_{1234} = \frac{1}{2}\Big( \langle \tau_1\tau_1\tau_2\tau_2 \rangle + \langle \tau_1\tau_2\tau_1\tau_2 \rangle + \langle \tau_1\tau_2\tau_2\tau_1 \rangle - \langle \tau_1\tau_1\tau_1\tau_1 \rangle \Big). \tag{139}$$

From the asymptotic behaviour (48) of $P_0 = P_{1234}$, we can deduce the three-point connectivity $C = 2^{\frac{1}{4}}$. This exactly agrees with the conjecture (43).

The relation (138) between the four-point functions in the odd CFT and in the Ashkin–Teller model makes it tempting to identify the fields $\tau_1, \tau_2$ of the Ashkin–Teller model with the fields $V^D, V^N$ of the odd CFT. However, the identification fails at the level of three-point functions, as in particular $\langle \tau_1\tau_1\tau_1 \rangle = \langle \tau_2\tau_2\tau_2 \rangle = 0$. In Appendix C.3, we actually show that for $Q = 4$ (in contrast to $Q = 3$) the fields $V^D, V^N$ cannot be linear combinations of the spins $\sigma_\alpha$. This is consistent with the fact that $\langle \tau_1\tau_1\tau_1\tau_1 \rangle$ has a discrete spectrum $\mathcal{S}_{2\mathbb{Z},\mathbb{Z}}$ [32], while we expect that $\langle V^D V^D V^D V^D \rangle$ has a continuous spectrum, as follows from taking the limit $c \to 1$ in minimal models. The odd CFT and the Ashkin–Teller model therefore manage to have some identical four-point functions, while being different CFTs.

# A    Connectivities and their scaling limits

In this section we explain how we deduce the scaling limits (44) from Monte-Carlo computations on finite lattices.

## A.1    Two-point connectivity

On a doubly periodic square lattice of size $L$, we measure the probability that two lattice points are in the same cluster. We take the pair of lattice points $(z_1, z_2)$ of type $(z_1, z_1 + r)$ or $(z_1, z_1 + ir)$, and the associated probability is a function $p_{12}(r, L)$ of $r$ and $L$. We compute the probability for a sample of $10^6$ graphs $\mathcal{G}$ (see Section 3), and we average over all possible positions of $z_1$ – so the effective sample size is $10^6 L^2$.

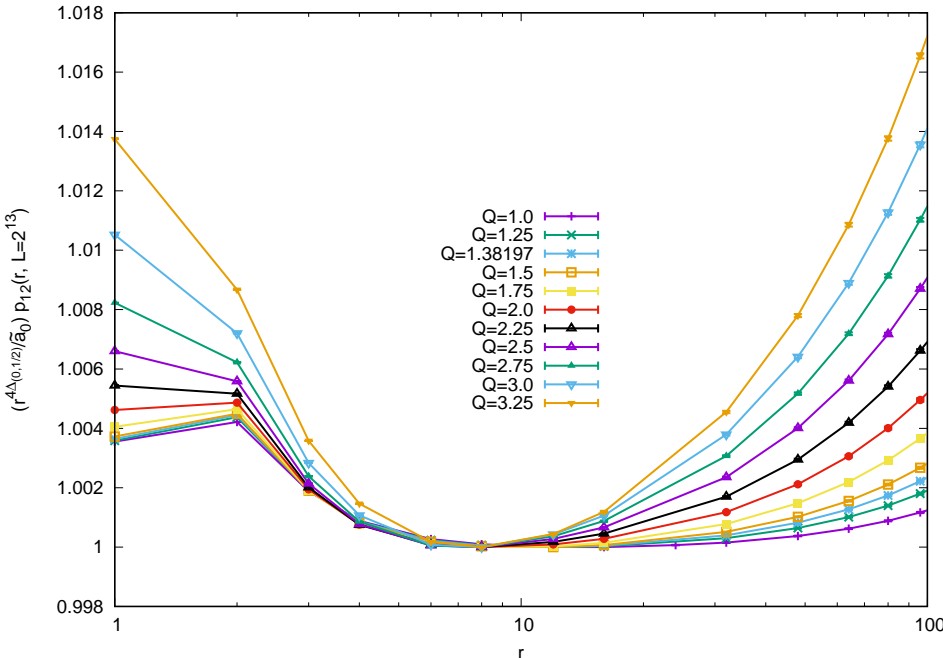

Figure A.1: Monte-Carlo measurements of the two-point connectivity as a function of the distance $r$ between the two points. The plot shows the function $\frac{1}{\tilde{a}_0} r^{4\Delta_{(0,\frac{1}{2})}} p_{12}(r, L = 2^{13})$, for $r = 1, 2, 4, 8, \cdots, \frac{L}{2}$ and $r = 3, 6, 12, 48, 80, 96$. $\tilde{a}_0$, defined in the text, takes values close to the $a_0$ of Table A.3.

According to Eq. (40), the quantity $r^{4\Delta_{(0,\frac{1}{2})}} p_{12}(r, L)$ should be constant in the scaling limit. In Figure A.1, we plot this quantity as a function of $r$ for $L = 2^{13}$ for different values of $Q$, after dividing by the normalization factor $\tilde{a}_0(Q) = \min_r \left( r^{4\Delta_{(0,\frac{1}{2})}} p_{12}(r, L) \right)$.

We observe an interplay between lattice and topological effects: when $r$ is of of the order of a hundred lattice sites $r = O(10^2)$, the probability deviates from its scaling limit due to the periodic boundary conditions. Lattices effect are visible when $r$ is a few lattice sites $r = O(1)$. In order to disentangle the lattice from the topological effects, we did the same plot for three different values of $L$ (in the case $Q = 2$) in Figure A.2. The lattice corrections that appear for $r = O(1)$ are $L$-independent, while the topological corrections become $L$-independent if the connectivity is rewritten as a function of $\frac{r}{L}$ as in the right part of Figure A.2.

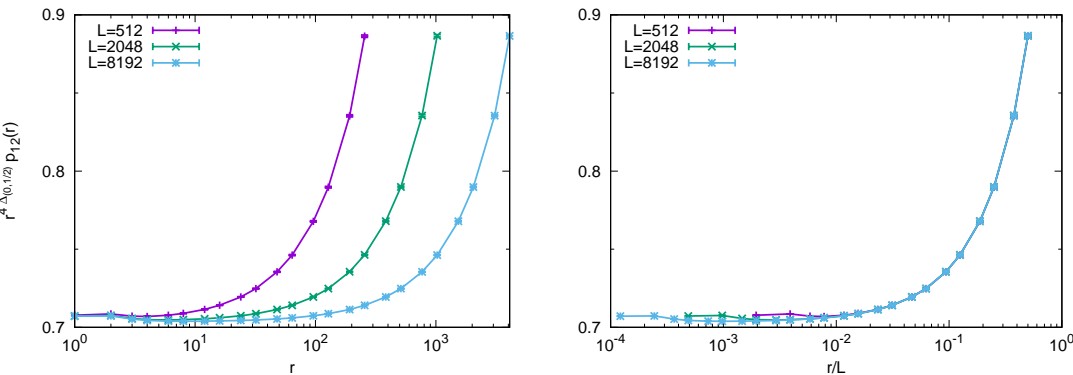

Figure A.2: Two-point connectivity of the Ising model $Q = 2$ for three values of the lattice size $L$. The plots show $r^{4\Delta_{(0,\frac{1}{2})}} p_{12}(r, L)$ as a function of $r$ (left) and $\frac{r}{L}$ (right).

Consistently with these numerical observations, and analogously to the case of four-point connectivities (58), we describe the lattice and topological corrections by the ansatz

$$p_{12}(r, L) = \frac{a_0}{r^{4\Delta_{(0,\frac{1}{2})}}} \left(1 + \alpha_1 \left(\frac{r}{L}\right)^{2\Delta_{(1,2)}}\right) \left(1 + \frac{d_1}{r^{d_2}}\right) . \tag{140}$$

(For further details on the validity of this ansatz, and a comparison to predictions from CFT on the torus, see [33].) We determine the parameters $a_0, \alpha_1, d_1, d_2$ by fitting our data to this ansatz. The quality of our fits is tested using the goodness of fit, which we denote as $\chi^2/ndf$. We find that the fit is very good for distances $r \in [6, \frac{L}{4}]$, see Table A.3, for the resulting values of the parameters.

Table A.3: Parameters of the ansatz (140) for the two-point connectivity, determined by fitting Monte-Carlo results.

| $Q$ | $a_0$ | $\alpha_1$ | $d_1$ | $d_2$ |
|---|---|---|---|---|
| 1 | 0.74719 | 0.356 | 0.016 | 2.03 (2) |
| 1.25 | 0.73323 | 0.392 | 0.018 | 2.07 (3) |
| $2 + 2\cos 3\pi/5$ | 0.72693 | 0.414 | 0.018 | 2.05 (2) |
| 1.5 | 0.72178 | 0.4343 | 0.021 | 2.15 (2) |
| 1.75 | 0.71199 | 0.459 | 0.019 | 2.09 (4) |
| 2.0 | 0.70337 | 0.488 | 0.019 (2) | 2.02 (6) |
| 2.25 | 0.69556 | 0.518 | 0.024 (2) | 2.12 (5) |
| 2.5 | 0.68827 | 0.551 | 0.029 (2) | 2.11 (4) |
| 2.75 | 0.68113 | 0.578 | 0.023 (2) | 1.76 (4) |
| 3.0 | 0.67376 (2) | 0.599 | 0.018 | 1.31 (4) |
| 3.25 | 0.66555 (5) | 0.627 | 0.019 | 0.97 (3) |
| $2 + \sqrt{2}$ | 0.65902 (7) | 0.642 | 0.022 | 0.78 (3) |

Let us further justify the ansatz (140) by comparing it with the known exact behaviour of the two-point connectivity in the Ising model. In the Ising model i.e. for $Q = 2$, the two-point connectivity coincides with a two-point function of spin observables. (See Section 4.3.) This two-point function has been computed exactly on an infinite square lattice [34], and its large $r$ behaviour is of the type

$$p_{12}(r, L = \infty) \underset{\substack{Q=2 \\ r \gg 1}}{=} \frac{a_0}{r^{\frac{1}{4}}} \left(1 + \frac{d_1}{r^2} + \cdots\right) , \tag{141}$$

with the parameters

$$a_0 \underset{Q=2}{=} 0.703380157 \quad , \quad d_1 \underset{Q=2}{=} \frac{1}{64} . \tag{142}$$

This agrees not only with the values for the parameters of our ansatz, but also with the direct numerical study of spin-spin lattice corrections in the Ising model [35]. When it comes to topological corrections, let us compare our ansatz (140) with the exact torus two-point functions that are known in the cases $Q = 2$ [28] and $Q = 4$ [36]. In both cases, the leading correction comes from a diagonal primary field $(1, 2)^D$. And for our square torus $\frac{\mathbb{C}}{L\mathbb{Z}+iL\mathbb{Z}}$, the parameter $\alpha_1$ at $Q = 2$ takes the value

$$\alpha_1 \underset{Q=2}{=} 0.488863 , \tag{143}$$

that agrees very well with our value of $\alpha_1$, see Table A.3. In the case $Q = 3$ too, the operator algebra and $\mathbb{Z}_3$ symmetry imply that the leading topological correction is given by the energy

field. Let us give numerical evidence for this observation in the cases $Q = 1, 2, 3$, by directly plotting the difference $r^{4\Delta_{(0,\frac{1}{2})}}p_{12}(r, L) - a_0$ in the range $r \in [50 : 200]$ where lattice corrections are negligible. This is done in Figure A.4, and the results are compatible with the values $2\Delta_{(1,2)} = \frac{5}{4}, 1, \frac{4}{5}$ for $Q = 1, 2, 3$ respectively.

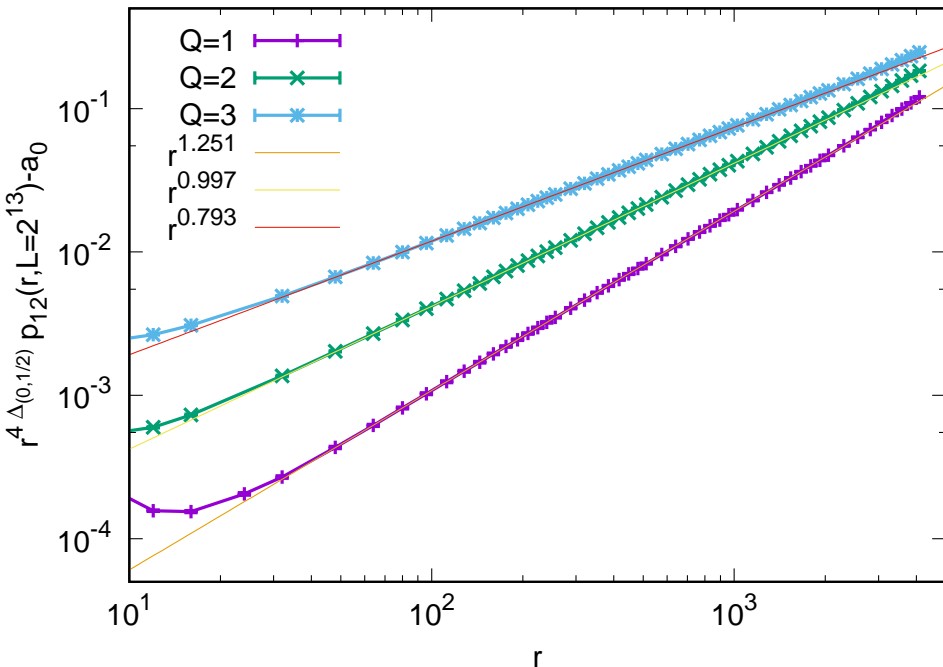

Figure A.4: Numerical results for $r^{4\Delta_{(0,\frac{1}{2})}}p_{12}(r, L = 2^{13}) - a_0$ as a function of $r$ in the cases $Q = 1, 2, 3$, and the best power law fits.

Table A.5: Parameters of the fit (144) in the case of the Ising model ($Q = 2$), as functions of the lattice size $L$.

| $L$ | 512 | 1024 | 2048 | 4096 | 8192 |
|---|---|---|---|---|---|
| $a_0$ | 0.7031 (2) | 0.70313 (7) | 0.70333 (2) | 0.70334 (2) | 0.70337 (1) |
| $\alpha_1$ | 0.491 (2) | 0.487 (2) | 0.4899 (7) | 0.4834 (8) | 0.4882 (7) |
| $d_1$ | 0.005 (3) | 0.006 (3) | 0.015 (4) | 0.016 (4) | 0.020 (3) |
| $d_2$ | 1.0 (4) | 1.2 (3) | 1.9 (2) | 1.9 (2) | 2.1 (1) |
| $\chi^2/ndf$ | 0.06 | 0.19 | 0.111 | 0.27 | 0.26 |

The first subleading topological correction already provides a very good approximation to the exact results at $Q = 2$ and this is due to the fact that the next topological correction are much smaller. The exact torus two-point function of [28] indeed leads to topological corrections of the type

$$p_{12}(r, L) \underset{\substack{Q=2 \\ 1 \ll r \ll L}}{=} \frac{a_0}{r^{\frac{1}{4}}}\left(1 + \alpha_1 \frac{r}{L} + \alpha_2 \left(\frac{r}{L}\right)^4 + \cdots\right), \tag{144}$$

where $\alpha_2 = 0.211556$. In particular, the term in $\left(\frac{r}{L}\right)^2$ vanishes in the case of the square torus. The term in $\left(\frac{r}{L}\right)^3$ would be proportional to the torus one-point function of the derivative of the energy-momentum tensor, and therefore vanishes. The next to leading correction is therefore in $\left(\frac{r}{L}\right)^4$: for this to be negligible, it is enough to require $r < \frac{L}{4}$. As shown in Table A.5,

the quality of the fit improves when $L$ increases, and when we reach $L = 2^{13} = 8192$ the numerically determined values of the parameters are compatible with the exact values.

## A.2 Four-point connectivities

In order to extract the scaling limit (44) we use the ansatz (58). In Figure A.6, we display the $r$-dependence of $p_{1234}, p_{12,34}, p_{13,24}, p_{14,23}$, in the case $Q = 2$.

From the Figure A.6, we observe that the large $r$ corrections are more regular for $p_{1234}$ and $p_{13;24}$ than for $p_{12;34}$ and $p_{14;23}$. These corrections increase when $z$ becomes smaller i.e. when our rectangle (55) becomes more elongated. In order to achieve a good fit with our ansatz, we focus on values of $r$ such that the corrections are not too strong. For example, in the case $z = 0.01538$, we focus on $r < 256$. More generally, the upper bound for $r$ comes from comparing the length $\lambda r$ of our rectangle with the size $L$ of the lattice, and we focus on $r \in [6, \frac{L}{4\lambda}]$. In this interval, our ansatz fits the numerical data very well, as we demonstrate in Figure A.7. In that Figure, we observe that the leading correction to the four-point connectivity is governed by the exponent $2\Delta_{(1,2)} = \frac{5}{4}, 1, \frac{4}{5}$ for $Q = 1, 2, 3$ respectively, as predicted by the fit (58). This generalizes to four-point connectivities what we already observed for two-point connectivities in Figure A.4.

## A.3 Numerical errors and their dependence on $Q$

Let us evaluate the numerical errors in our Monte-Carlo measurements of connectivities. To do this, we focus on quantities for which exact analytic results are known. We start with the analytically known four-point connectivities for $Q = 2$ and $Q = 3$.

In the case $Q = 2$, let us test the simple exact result (118) for the sum of the four connectivities (116). We plot the ratio between the measurements and the exact result on the left part of Figure A.8, and find a relative error of the order $10^{-4}$. On the right part of the figure, we plot the first subleading topological correction $c_1(z)$ (115), compared to the analytic result (119), and we find a relative error of the order $10^{-2}$.

In the case $Q = 3$, out of the three exactly known combinations of connectivities, we choose the combination

$$\left\langle V^D V^D V^D V^D \right\rangle = R_1 + R_2 + R_3 = \frac{3}{2} P_0 + P_1 + P_2 + P_3 \,, \tag{145}$$

whose expression is given in Eq. (132). In Figure A.9, we plot the relative error between the analytic expression, and Monte-Carlo measurements done with three different choices of parameters: $L = 2^{11}$, $L = 2^{13}$, and $L = 2^{13}$ with subleading topological corrections factored in. The relative error is of the order $10^{-3}$ in the latter case.

Therefore, we find numerical errors of the order $10^{-4}$ for $Q = 2$ and $10^{-3}$ for $Q = 3$. This suggests that numerical errors are strongly increasing functions of $Q$. This explains why the difference between the measured connectivities and the odd CFT's four-point functions increases with $Q$, even though the odd CFT gives exact results at $Q = 3$ and $Q = 4$.

# B Extracting the spectrum and the structure constants

In this section we give details on the extraction of the small $z$ asymptotics of $P_\sigma(z)$ from Monte-Carlo measurements. This is the technical basis for the results of Section 3.4.

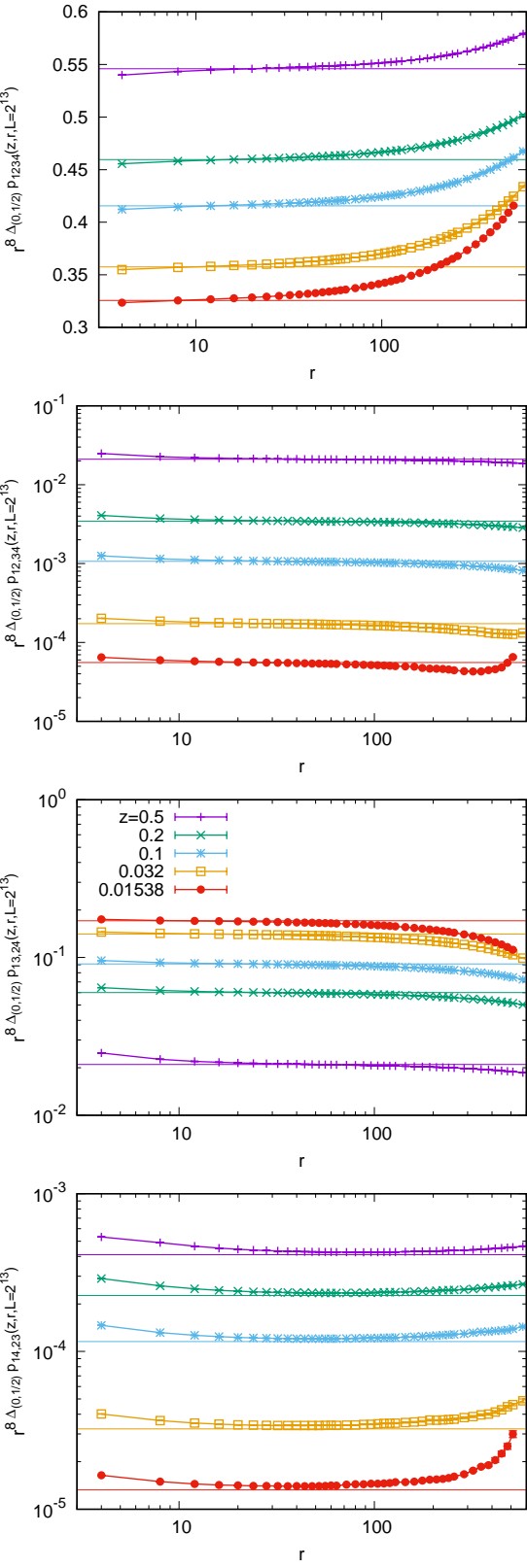

Figure A.6: $r^{8\Delta_{(0,\frac{1}{2})}} p_X(z|r, L = 2^{13})$ as functions of $r$ for various cross-ratios $z$, in the case $Q = 2$. Clockwise from the upper left panel, we display $X = 1234$, then $12, 34$, then $14, 23$, then $13, 24$.

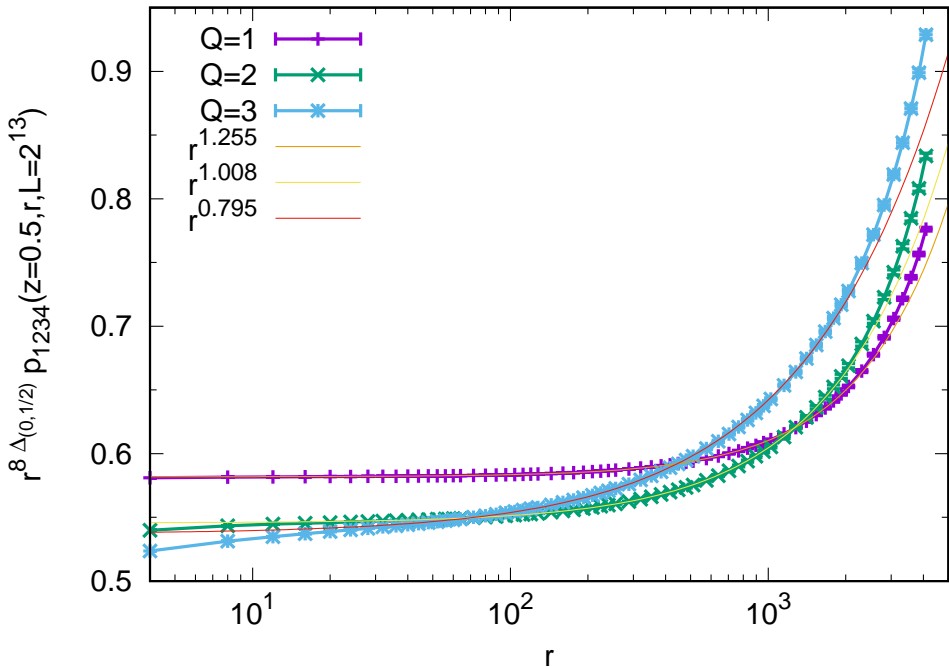

Figure A.7: Rescaled four-point correlation function $r^{8\Delta_{(0,\frac{1}{2})}}p_{1234}(z=\frac{1}{2}|r, L=2^{13})$ for $Q=1,2,3$ as a function of $r$.

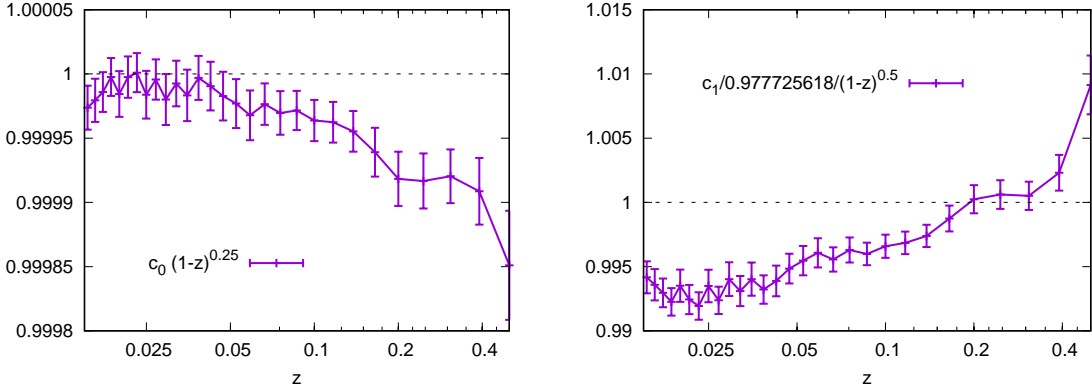

Figure A.8: $Q=2$ comparison between Monte-Carlo measurements and exact results for leading (left) and subleading (right) contributions to four-point connectivities. We plot ratios between measurements and exact results, as functions of the cross-ratio $z$.

## B.1 Principles

The general idea is to fit Monte-Carlo data with a truncated $s$-channel conformal block expansion (66), where we want to determine the spectrum $\{\Delta, \bar{\Delta}\}$ and structure constants $D_{\Delta,\bar{\Delta}}$. We either take values of $\Delta, \bar{\Delta}$ from the exact predictions (27), or leave them as parameters to be fitted and compared to the predictions. Similarly, for the structure constants, we sometimes take exact values in cases when we know them, or leave them as parameters.

When $\Delta, \bar{\Delta}$ are considered as given, we compute the corresponding conformal blocks using Zamolodchikov's recursion. When however we want to fit the conformal dimension, we approximate the block with the first two terms of its small $z$ expansion. Moreover, we only fit

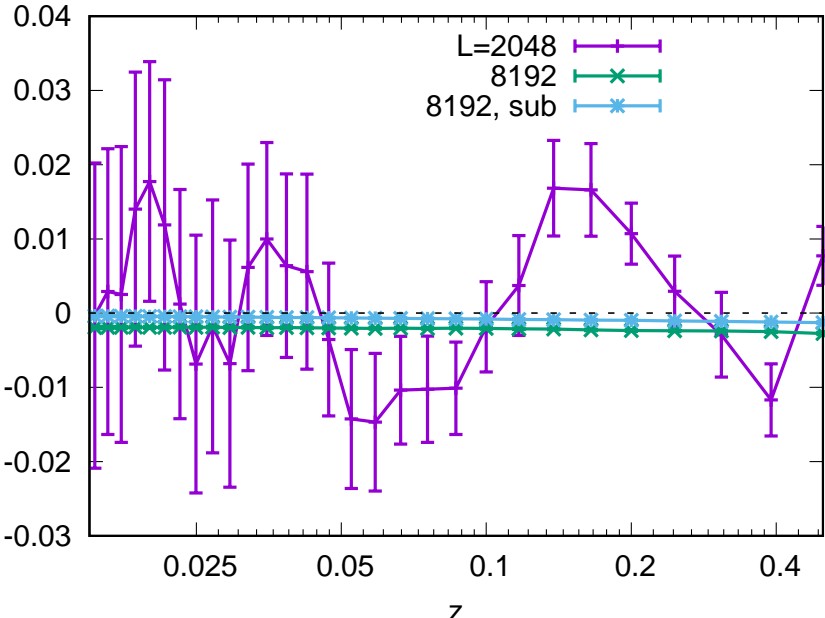

Figure A.9: $Q = 3$ comparison between Monte-Carlo measurements and exact results for $L = 2^{11}$, $L = 2^{13}$, and $L = 2^{13}$ with subleading topological corrections factored in. Relative differences are plotted as functions the cross-ratio.

data for real positive $z$: if $\beta = \Delta + \bar{\Delta}$ is the total conformal dimension, then our blocks look like

$$\mathcal{F}^{(s)}_{\Delta}(z)\mathcal{F}^{(s)}_{\bar{\Delta}}(\bar{z}) \underset{z>0}{=} z^{\beta}\left(1 + \tfrac{1}{2}\beta z + O(z^2)\right) . \tag{146}$$

After fitting the parameters, we can check that the fit also matches some Monte-Carlo data for complex $z$. For general complex $z = \rho e^{i\theta}$, we write the conformal dimensions as $(\Delta, \bar{\Delta}) = (\tfrac{\beta+S}{2}, \tfrac{\beta-S}{2})$ where $S \in \mathbb{Z}$ is the conformal spin, and the blocks behave as

$$\mathcal{F}^{(s)}_{\Delta}(z)\mathcal{F}^{(s)}_{\bar{\Delta}}(\bar{z}) + \mathcal{F}^{(s)}_{\Delta}(\bar{z})\mathcal{F}^{(s)}_{\bar{\Delta}}(z)$$
$$\underset{z=\rho e^{i\theta}}{=} 2\rho^{\beta}\cos S\theta + \beta\rho^{\beta+1}\cos\theta\cos S\theta - S\rho^{\beta+1}\sin\theta\sin S\theta + O(\rho^{\beta+2}) . \tag{147}$$

For example, for a state of spin $S = 1$ like $(2, \tfrac{1}{2})$, the leading term vanishes for $z \in i\mathbb{R}$, and the blocks are dominated by a term of order $O(\rho^{\beta+1})$.

In order for the fits to be reasonably significant, we attempt to fit at most 4 parameters. We measure the goodness of fit by the quantity chi squared over the number of degrees of freedom ($\chi^2/ndf$): the lower this quantity, the better the fit, with good fits having values of order one or less. The goodness of fits depends not only on which parameters we include, but also on the choice of the values of $z$. We performed our fits on the interval $z \in (0, 0.25)$, where we found that the first few terms of the small $z$ asymptotic expansion describe the four-point connectivities very well.

## B.2 Fitting $P_0$ and $P_1$

Let us fit $P_0$ and $P_1$ in order to check the predicted spectrums (27), and to test the behaviour of $P_0 + P_1$ that we predicted in (52). We us the following four-parameter ansatzes:

$$P_0(z) \sim \alpha_0^{(1)} z^{\beta_0^{(1)}} \left(1 + \tfrac{1}{2}\beta_0^{(1)} z\right) + \alpha_0^{(2)} z^{\beta_0^{(2)}} \left(1 + \tfrac{1}{2}\beta_0^{(2)} z\right), \tag{148}$$

$$P_1(z) \sim 1 + \alpha_1^{(1)} z^{\beta_1^{(1)}} \left(1 + \tfrac{1}{2}\beta_1^{(1)} z\right) + \alpha_1^{(2)} z^{\beta_0^{(2)}} \left(1 + \tfrac{1}{2}\beta_1^{(2)} z\right). \tag{149}$$

The predictions that we are testing are

$$\beta_0^{(1)} = \beta_1^{(1)} = 2\Delta_{(0,\frac{1}{2})}, \quad \begin{cases} \beta_0^{(2)} \underset{Q\leq 2}{=} 2\Delta_{(2,0)} \\ \beta_0^{(2)} \underset{Q\geq 2}{=} 2\Delta_{(0,\frac{3}{2})} \end{cases}, \quad \beta_1^{(2)} = 2\Delta_{(1,0)}, \tag{150}$$

for the conformal dimensions, and

$$\alpha_0^{(1)} = -\alpha_1^{(1)} = 2D_{(0,\frac{1}{2})}, \tag{151}$$

for the structure constants. We display the results in Table B.1. They agree well with our predictions for $\beta_0^{(1)}, \beta_1^{(1)}, \alpha_0^{(1)}$ and $\alpha_1^{(1)}$. We observe the usual decrease of precision when $Q$ increases, which we attribute to numerical artefacts. (Our predictions are exactly verified for $Q = 4$.) The agreement is not so good for $\beta_0^{(2)}$ and $\beta_1^{(2)}$, which is not too surprising since our ansatz neglects contributions whose conformal dimensions are not far above the predicted values. The values of $\beta_0^{(2)}$ and $\beta_1^{(2)}$ are also much less robust with respect to changes in the fit range $z \in (0, 0.25)$. We plot the resulting fits for $P_0(z)$ in Figure B.2, together with the Monte-Carlo

Table B.1: Fit to the form (148) of $P_0(z)$ and $P_1(z)$. We show error bars in parentheses when the error exceeds one unit of the last printed digit.

| $Q$ | $2\Delta_{(0,\frac{1}{2})}$ | $2D_{(0,\frac{1}{2})}$ | $\alpha_0^{(1)}$ | $\beta_0^{(1)}$ | $\beta_0^{(2)}$ | $\alpha_1^{(1)}$ | $\beta_1^{(1)}$ | $\beta_1^{(2)}$ |
|---|---|---|---|---|---|---|---|---|
| 1 | 0.10417 | 1.0445 | 1.046 | 0.1045 | 1.42 | -1.051 | 0.1053 | 0.84 |
| 1.25 | 0.11118 | 1.0588 | 1.060 | 0.1113 | 1.42 | -1.067 | 0.1123 | 0.83 |
| 1.382 | 0.1143 | 1.0668 | 1.068 | 0.1146 | 1.43 | -1.076 | 0.1157 | 0.82 |
| 1.5 | 0.11678 | 1.0741 | 1.075 | 0.1170 | 1.44 | -1.085 | 0.1185 | 0.79 |
| 1.75 | 0.12131 | 1.0903 | 1.092 | 0.1218 | 1.49 | -1.107 | 0.1239 | 0.75 |
| 2.0 | 0.125 | 1.1077 | 1.110 | 0.1254 | 1.50 (2) | -1.128 | 0.1279 (2) | 0.73 |
| 2.25 | 0.12798 | 1.1263 | 1.128 | 0.1283 | 1.46 (2) | -1.149 (2) | 0.1311 (2) | 0.70 |
| 2.5 | 0.13034 | 1.1464 | 1.148 | 0.1308 | 1.48 (2) | -1.178 (2) | 0.1345 (3) | 0.66 |
| 2.75 | 0.13212 | 1.1686 | 1.170 | 0.1326 | 1.45 (2) | -1.212 (3) | 0.1377 (4) | 0.62 |
| 3.0 | 0.13333 | 1.1933 | 1.194 | 0.1339 | 1.47 (3) | -1.247 (4) | 0.1399 (5) | 0.58 |
| 3.25 | 0.13393 | 1.22157 | 1.224 | 0.1353 | 1.68 (4) | -1.306 (4) | 0.1438 (4) | 0.53 |
| 3.414 | 0.13393 | 1.2431 | 1.242 | 0.1354 | 1.52 (5) | -1.349 (7) | 0.1461 (6) | 0.50 |

data. We see that our fits are notably better than the first term truncation $2D_{(0,\frac{1}{2})} \left| \mathcal{F}_{\Delta_{(0,\frac{1}{2})}}^{(s)}(z) \right|^2$. If we then set $\beta_0^{(1)} = \beta_1^{(1)} = 2\Delta_{(0,\frac{1}{2})}$ instead of having $\beta_0^{(1)}, \beta_1^{(1)}$ as parameters, and use an accurate computation of the corresponding conformal block $\mathcal{F}_{\Delta_{(0,\frac{1}{2})}}^{(s)}$, we obtain three-parameter fits whose results agree with the predictions $\alpha_0^{(1)} = -\alpha_1^{(1)} = 2D_{(0,\frac{1}{2})}$ even better.

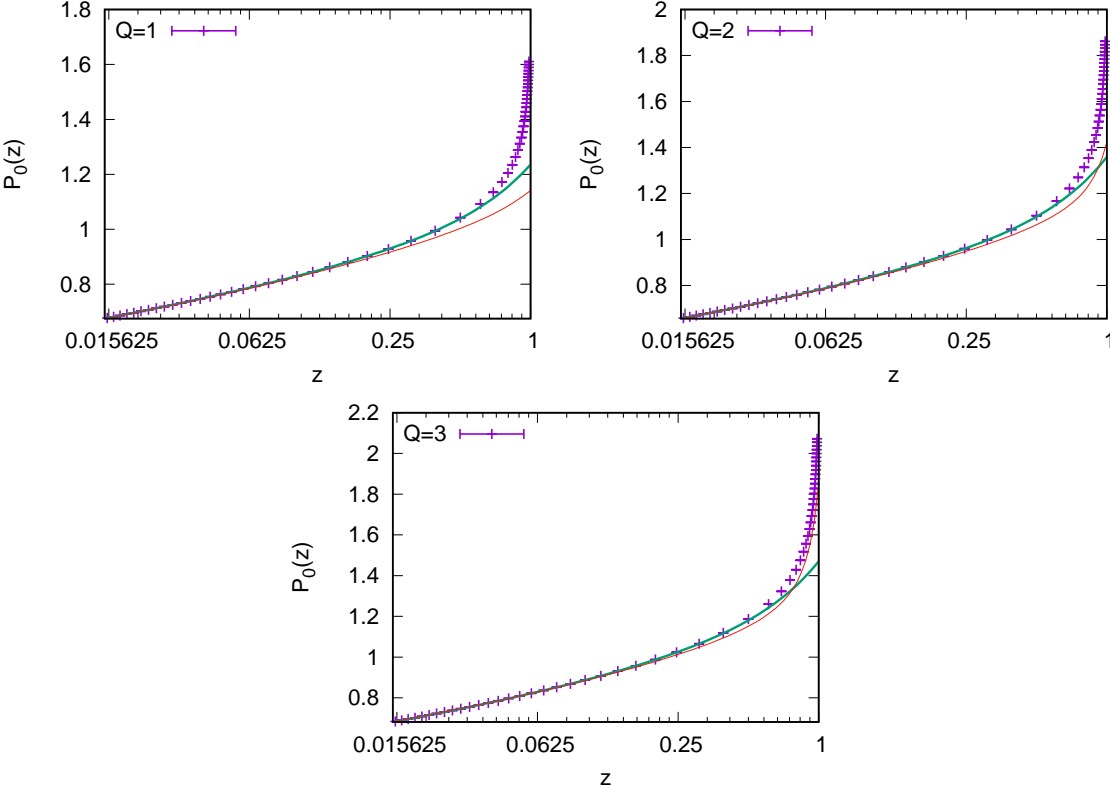

Figure B.2: $P_0(z)$ as a function of the cross-ratio $z$ for $Q = 1, 2, 3$: comparing Monte-Carlo data (crosses) with the first term truncation (thin red line) and the best fit of the form (148) (blue line).

In order to get a reasonably precise determination of the structure constant of the state $(2, 0)$ in $P_0$, let us introduce an ansatz where this structure constant is the sole parameter $\alpha_0^{(2)}$:

$$P_0(z) = D_{(0,\frac{1}{2})} \left| \mathcal{F}^{(s)}_{\Delta_{(0,\frac{1}{2})}}(z) \right|^2 + D_{(0,\frac{3}{2})} \left| \mathcal{F}^{(s)}_{\Delta_{(0,\frac{3}{2})}} \right|^2 + \alpha_0^{(2)} \left| \mathcal{F}^{(s)}_{\Delta_{(2,0)}} \right|^2 . \tag{152}$$

(In order for this equation to be consistent with the notations of Eq. (67), $\alpha_0^{(2)}$ should be replaced with $\alpha_0^{(3)}$ if $Q > 2$.) In the case $Q = 1$, we find the value

$$\alpha_0^{(2)} \underset{Q=1}{=} 0.0718(2) . \tag{153}$$

This allows us to test the prediction (70) were $\alpha_{2,3}^{(1)}$ will be given in Tables B.3 and B.4. We find that the prediction holds quite well.

## B.3 Fitting $P_2$ and $P_3$

Let us study $P_2$ and $P_3$, and in particular the four lowest-lying states (27). We assume that the structure constants of the odd spin states $(2, \frac{1}{2})$ and $(2, \frac{3}{2})$ are accurately given by the relation (61) with the odd CFT correlation functions $R_2, R_3$. We also assume that the ground state is $(2, 0)$. We however do not assume the state $(2, 1)$ to be present, but allow a state with an

Table B.3: Numerical results for the parameters of the ansatz (154) in the case of $P_2$.

| $Q$ | $\alpha_2^{(1)}$ | $\alpha_2^{(3)}$ | $\beta_2^{(3)}$ | $\Delta_{(2,1)} + \Delta_{(2,-1)}$ | $\chi^2/ndf$ |
|---|---|---|---|---|---|
| 1 | 0.03467(2) | 0.0089 | 2.09(2) | 2 | 5.1 |
| 1.25 | 0.03408(2) | 0.0097 | 2.14 | 2.03323 | 3.0 |
| 1.38 | 0.03366(2) | 0.0100 | 2.17 | 2.05 | 1.0 |
| 1.5 | 0.03306(2) | 0.0101 | 2.17 | 2.06468 | 3.9 |
| 1.75 | 0.03157(2) | 0.0103 | 2.18 | 2.09508 | 2.3 |
| 2.25 | 0.02811(3) | 0.0103 | 2.25 | 2.15492 | 3.3 |
| 2.5 | 0.02641(4) | 0.0103 | 2.35 | 2.18532 | 2.2 |
| 2.75 | 0.02408(4) | 0.0098 | 2.32(2) | 2.21677 | 4.5 |
| 3.0 | 0.02240(3) | 0.0093 | 2.47 | 2.25 | 1.4 |

Table B.4: Numerical results for the parameters of the ansatz (154) in the case of $P_3$.

| $Q$ | $\alpha_3^{(1)}$ | $\alpha_3^{(3)}$ | $\beta_3^{(3)}$ | $\chi^2/ndf$ |
|---|---|---|---|---|
| 1 | 0.03440(3) | 0.0088 | 2.03 | 0.48 |
| 1.25 | 0.03404(2) | 0.0098 | 2.17 | 0.36 |
| 1.38 | 0.03346(2) | 0.0099 | 2.15 | 0.25 |
| 1.5 | 0.03257(3) | 0.0096 | 2.05 | 1.0 |
| 1.75 | 0.03132(3) | 0.0103 | 2.15 | 0.50 |
| 2.25 | 0.02803(6) | 0.0103 | 2.24 | 1.16 |
| 2.5 | 0.02650(4) | 0.0100(2) | 2.34(2) | 0.71 |
| 2.75 | 0.02475(7) | 0.0102(3) | 2.44(4) | 1.15 |
| 3.0 | 0.02276(3) | 0.0096 | 2.43 | 0.11 |

undetermined dimension instead. This leads to the three-parameter ansatzes

$$
P_\sigma(z) \underset{z>0}{=} (-1)^\sigma (Q-2) \left[ 2D_{(2,\frac{1}{2})} \mathcal{F}^{(s)}_{\Delta_{(2,\frac{1}{2})}}(z) \mathcal{F}^{(s)}_{\Delta_{(2,-\frac{1}{2})}}(z) + 2D_{(2,\frac{3}{2})} \mathcal{F}^{(s)}_{\Delta_{(2,\frac{3}{2})}}(z) \mathcal{F}^{(s)}_{\Delta_{(2,-\frac{3}{2})}}(z) \right]
$$
$$
+ \alpha_\sigma^{(1)} \left| \mathcal{F}^{(s)}_{\Delta_{(2,0)}}(z) \right|^2 + \alpha_\sigma^{(3)} z^{\beta_\sigma^{(3)}} \left(1 + \tfrac{1}{2}\beta_\sigma^{(3)} z\right) , \qquad (\sigma = 2,3) . \quad (154)
$$

The numerical results for the ansatzes' parameters are displayed in Tables B.3 and B.4, and they agree well with the predictions

$$
\alpha_2^{(1)} = \alpha_3^{(1)} , \; \alpha_2^{(3)} = \alpha_3^{(3)} , \; \beta_2^{(3)} = \beta_3^{(3)} = \Delta_{(2,1)} + \Delta_{(2,-1)} . \quad (155)
$$

The agreement is not particularly good for $\beta_2^{(3)}$ and $\beta_3^{(3)}$, which always exceed the predicted values: this is a hint that we are neglecting the contributions of states with higher conformal dimensions. Moreover, the goodness of fits is much better for $P_3$ than $P_2$. This is because the connectivity $P_2$ is much smaller than the other connectivities, and therefore determined with much less precision. (See Figure 3.5.)

## B.4 An excursion in the complex plane

Out of pure convenience, we have done most Monte-Carlo calculations for cross-ratios $z \in (0, 1)$. However, connectivities are defined for all $z \in \mathbb{C}$. When it comes to $z \to 0$ expansions, our simplified formula (146) for conformal blocks depends only on the total conformal dimension, and not on the conformal spin. However, for complex values of $z$, the behaviour of conformal

blocks strongly depends on the spin, including at the leading order, see Eq. (147). Therefore, comparing our fits with Monte-Carlo data for complex values of $z$ is a strong independent test of their validity, and of the predicted values of the spins.

Let us perform the comparison in the case $Q = 1$. We display Monte-Carlo data for $z = 0.02$ and $z = i0.02$, together with our fit's values for $z = i0.02$:

| | $z$ | $P_0$ | $P_1$ | $P_2$ | $P_3$ | |
|---|---|---|---|---|---|---|
| Monte-Carlo | 0.02 | 0.69616(3) | 0.30594(3) | 0.0001188(2) | 0.0004126(4) | |
| Monte-Carlo | $i0.02$ | 0.69552(2) | 0.30652(2) | 0.0002546(2) | 0.0002578(3) | (156) |
| Fit | $i0.02$ | 0.69541 | 0.30620 | 0.0002596 | 0.0002540 | |

We observe that $P_0(z)$ and $P_1(z)$ change only weakly as $z$ rotates from $0.02$ to $i0.02$: this is because they are dominated by states with zero spin. On the other hand, $P_2(z)$ and $P_3(z)$ change a lot, because they involve large contributions from the states with odd spins $(2, \frac{1}{2})$ and $(2, \frac{3}{2})$. In any case, the Monte-Carlo data agrees well with our fits for $z = i0.02$, even though the fits' parameters were determined using real values of $z$. In particular, since the leading terms of the conformal blocks for the states $(2, \frac{1}{2})$ and $(2, \frac{3}{2})$ vanish at $z = i0.02$, we expect that $P_2(i0.02)$ and $P_3(i0.02)$ are well-approximated by the sole contribution of the state $(2, 0)$. This contribution is $\alpha_2^{(1)}(0.02)^{\frac{5}{4}} = 0.0002595$ (with $\alpha_2^{(1)} \simeq 0.0345$), which agrees well with $P_2(i0.02) \simeq P_3(i0.02)$.

## C  Special cases: supplementary material

### C.1  Asymptotics of spanning trees

The determinants in (77) can be expressed in terms of the Gaussian integrals of Grassmann variables. Let $\psi_{z_i}$ and $\bar{\psi}_{z_i}$, be Grassmann numbers associated to a point $z_i$ of the square lattice. One can show that:

$$\det{}'[A] = \int \left( \prod_{z_i} d\psi_{z_i} \prod_{z_j} d\bar{\psi}_{z_j} \right) \psi_{z_l} \bar{\psi}_{z_l} e^{\sum_{z_i, z_j} \bar{\psi}_{z_j} A_{z_i z_j} \psi_{z_i}} \,, \tag{157}$$

$$\det\left[A(z_1, z_2; z_3, z_4)\right] = \int \left( \prod_{z_i} d\psi_{z_i} \prod_{z_j} d\bar{\psi}_{z_j} \right) \psi_{z_1} \psi_{z_2} \bar{\psi}_{z_3} \bar{\psi}_{z_4} e^{\sum_{z_i, z_j} \bar{\psi}_{z_j} A_{z_i z_j} \psi_{z_i}} \,. \tag{158}$$

To compute the correlation functions, one has to take into account that $A$ has a vanishing eigenvalue. Let us introduce a regulator $\lambda$ for the resulting divergences, and define

$$\langle \bar{\psi}_{z_1} \psi_{z_2} \rangle_\lambda = \frac{1}{\det[A + \lambda J]} \int \left( \prod_{z_i} d\psi_{z_i} \prod_{z_j} d\bar{\psi}_{z_j} \right) \bar{\psi}_{z_1} \psi_{z_2} \, e^{\sum_{i,j} \bar{\psi}_j (A_{z_i z_j} + \lambda J_{z_i z_j}) \psi_i} \tag{159}$$

$$= (A + \lambda J)^{-1}_{z_1 z_2} \,, \tag{160}$$

where $J$ is the $L^2 \times L^2$ matrix with all equal entries $J_{z_i, z_j} = L^{-2}$. The matrix $A + \lambda J$ has the eigenvalues

$$\left\{ 4 - 2\cos\left(\frac{2\pi}{L} n_x\right) - 2\cos\left(\frac{2\pi}{L} n_y\right) + \lambda \delta_{n_x, 0} \delta_{n_y, 0} \right\}_{n_x, n_y = 0, \cdots, L-1} \,. \tag{161}$$

The two-point function takes the form

$$\langle \bar{\psi}_{z_1} \psi_{z_2} \rangle_\lambda = \frac{1}{\lambda} + \frac{1}{L^2} \sum_{\substack{n_x, n_y = 0 \\ (n_x, n_y) \neq (0,0)}}^{L-1} \frac{e^{\frac{2\pi i}{L} n_x \Re z_{12} + \frac{2\pi i}{L} n_y \Im z_{12}}}{4 - 2\cos\left(\frac{2\pi}{L} n_x\right) - 2\cos\left(\frac{2\pi}{L} n_y\right)} \,. \tag{162}$$

Setting $z_1 = z_2$, we obtain

$$\lim_{\lambda \to 0} \det[A + \lambda J] \langle \bar{\psi}_{z_l} \psi_{z_l} \rangle_\lambda = \lim_{\lambda \to 0} \frac{1}{\lambda} \det[A + \lambda J] = {\det}'[A] \ , \tag{163}$$

consistently with Eq. (157). In the large distance limit, we have

$$\langle \bar{\psi}_{z_1} \psi_{z_2} \rangle_\lambda \underset{1 \ll |z_{12}| \ll L}{\sim} \frac{1}{\lambda} - \frac{1}{2\pi} \log\left( \frac{|z_{12}|}{L} \right) \ . \tag{164}$$

The logarithmic term can be interpreted as the fundamental solution of the Laplace equation $\Delta \frac{1}{2\pi} \log(|z|) = \delta^{(2)}(z)$, and indeed the discrete Laplacian $A$ tends to the Laplace operator $\Delta$ in the large distance limit. We finally compute

$$\frac{\det\left[ A(z_1, z_2; z_3, z_4) \right]}{{\det}'[A]} \underset{1 \ll |z_{12}| \ll L}{\sim} \lim_{\lambda \to 0} \lambda \left[ \langle \bar{\psi}_{z_1} \psi_{z_3} \rangle_\lambda \langle \bar{\psi}_{z_2} \psi_{z_4} \rangle_\lambda - \langle \bar{\psi}_{z_1} \psi_{z_4} \rangle_\lambda \langle \bar{\psi}_{z_2} \psi_{z_3} \rangle_\lambda \right] \tag{165}$$

$$= \frac{1}{4\pi} \left[ \log(1 - z) + \log(1 - \bar{z}) \right] \ , \tag{166}$$

which leads to Eq. (78). (Due to $\Delta_{(0, \frac{1}{2})} \underset{Q=0}{=} 0$, the prefactor in the definition (40) of the limit is constant.)

## C.2 $Q = 3$: correlation functions from the $\mathcal{W}_3$ algebra

Let us briefly review some basic formulas related to the $\mathcal{W}_3$ algebra. For the conventions and the notations used here, we refer the reader to [37] and references therein.

In the Toda parametrization, the central charge $c$ is given by

$$c = 2 + 24\left( b + \frac{1}{b} \right)^2 \ . \tag{167}$$

A primary field $V_\alpha$ can be labelled by a two component vector $\vec{\alpha} = (\alpha_1, \alpha_2)$. This can be understood from the fact the $\mathcal{W}_3$ theory admits a representations in terms of a two component bosonic field $\vec{\phi} = (\phi_1, \phi_2)$ and its vertex fields are $V_{\vec{\alpha}} = e^{\vec{\alpha}\vec{\phi}}$. The conformal dimension is:

$$\Delta_{\vec{\alpha}} = \frac{1}{2} \vec{\alpha} \left( 2\left( b + \frac{1}{b} \right)(\vec{\omega}_1 + \vec{\omega}_2) - \vec{\alpha} \right) \ .$$

In the above expression the $\vec{\omega}_i$, $i \in \{1, 2\}$, are the fundamental $sl_3$ weights. Given a set of positive integers, $r_1, r_2, s_1, s_2 \in \mathbb{Z}_+$, the field $V_{(r_1 r_2)|(s_1 s_2)}$ associated to the vector $\vec{\alpha}_{r_1 r_2, s_1 s_2}$

$$\vec{\alpha}_{r_1 r_2, s_1 s_2} = \left( b + \frac{1}{b} \right)(\vec{\omega}_1 + \vec{\omega}_2) + \sum_{i=1}^{2} \left( -b r_i - \frac{1}{b} s_i \right) \vec{\omega}_i \ , \tag{168}$$

is said to be fully degenerate: in its representation there are two null vectors, at level $r_1 s_1$ and $r_2 s_2$. Besides the identity, the fully degenerate primary with the lowest level null vectors, are associated to the fundamental representation:

$$V_{(2,1)|(1,1)}, \quad V_{(1,1)|(2,1)} \ , \tag{169}$$

and to the anti-fundamental one:

$$V_{(1,2)|(1,1)}, \quad V_{(1,1)|(1,2)} \ . \tag{170}$$

Let us consider the four point function:

$$\left\langle V_{\vec{\alpha}}(0) V_{(2,1)|(1,1)}(z) V_{s\vec{\omega}_1}(1) V_{\vec{\beta}}(\infty) \right\rangle , \tag{171}$$

where $s \in \mathbb{R}$ and the vectors $\alpha$ and $\beta$ are general. This function satisfies a third order differential equation whose solutions correspond to the three fusion channels:

$$V_{(2,1)|(1,1)} \otimes V_{\vec{\alpha}} = \sum_{i=1}^{3} V_{\vec{\alpha} - b\vec{h}_i} , \tag{172}$$

where $\vec{h}_i$ are the weights in the fundamental representation:

$$\vec{h}_1 = \vec{\omega}_1 \quad \vec{h}_2 = -\vec{\omega}_1 + \vec{\omega}_2 \quad \vec{h}_3 = -\vec{\omega}_2 . \tag{173}$$

The explicit form of the above correlation function has been computed in [38] and takes the form:

$$\left\langle V_{\vec{\alpha}}(0) V_{(2,1)|(1,1)}(z) V_{s\vec{\omega}_1}(1) V_{\vec{\beta}}(\infty) \right\rangle = \sum_{j=1}^{3} D_j \, |H_j(z)|^2 , \tag{174}$$

where:

$$H_1(z) = z^{-\frac{b^2}{3}} (1-z)^{b^2 - \frac{bs}{3} + 1} \, {}_3F_2(A_1, A_2, A_3; B_1, B_2; z) , \tag{175}$$

$$H_2(z) = z^{-B_1 + \frac{b^2}{3} + 2} (1-z)^{b^2 - \frac{bs}{3} + 1} \times \tag{176}$$

$$\times \, {}_3F_2(A_1 - B_1 + 1, A_2 - B_1 + 1, A_3 - B_1 + 1; 2 - B_1, -B_1 + B_2 + 1; z) , \tag{177}$$

$$H_3(z) = z^{-B_2 - \frac{b^2}{3} + 2} (1-z)^{b^2 - \frac{bs}{3} + 1} \times \tag{178}$$

$$\times \, {}_3F_2(A_1 - B_2 + 1, A_2 - B_2 + 1, A_3 - B_2 + 1; B_1 - B_2 + 1, 2 - B_2; z) , \tag{179}$$

with

$$P_{\vec{\alpha}} = \vec{\alpha} - \left( b + \frac{1}{b} \right)(\vec{\omega}_1 + \vec{\omega}_2) , \tag{180}$$

$$A_i = 1 - \frac{bs}{3} + \frac{b^2}{3} + b(P_{\vec{\alpha}} h_1 + P_{\vec{\beta}} . h_i) , \tag{181}$$

$$B_1 = 1 + b \, P_{\vec{\alpha}} . (2\omega_1 - \omega_2) \quad B_2 = 1 + b P_{\vec{\alpha}} (\omega_1 + \omega_2) . \tag{182}$$

The structure constants $D_j$ are:

$$D_k = \left[ \frac{1}{\gamma[b^2]^{k-1}} \prod_{i=1}^{k-1} \frac{\gamma\left[ bP_{\vec{\alpha}} . (\vec{h}_i - \vec{h}_k) \right]}{\gamma\left[ 1 + b^2 + bP_{\vec{\alpha}} . (\vec{h}_i - \vec{h}_k) \right]} \right]^2 . \tag{183}$$

The above results are true for general $c$. We are interested in specifing the above result for the $\mathcal{W}_3$ minimal models, where:

$$b = i\sqrt{\frac{p}{p'}} ,$$

with $p, p'$ positive coprime integers. The spectrum is composed by fully degenerate fields, the two sets integers satisfy the following constraints:

$$1 \le r_1 + r_2 \le p, \quad 1 \le s_1 + s_2 \le p' . \tag{184}$$

We want to compute:

$$\left\langle V_{(2,1)(1,1)}(0)V_{(1,2)(1,1)}(z)V_{(2,1)(1,1)}(1)V_{(1,2)(1,1)}(\infty)\right\rangle \tag{185}$$

for the $WA_3$ minimal model with central charge $c = \frac{4}{5}$ with $(p, p') = (4, 5)$. This is obtained by setting in (174):

$$b = i\sqrt{\frac{4}{5}}, \quad \vec{\alpha} = \vec{\beta} = -b\,\vec{\omega}_2, \quad s = -b \tag{186}$$

to In this case, the structure constants $D_2$ vanishes , leaving only two channels. In the $s-$ channel:

$$V_{(2,1)|(1,1)} \times V_{(1,2)|(1,1)} = V_{-b\omega_1 - b\,\omega_2}(= V_{22|11}) \oplus V_{-b\omega_2 - bh_3}(= \text{Id}) , \tag{187}$$

while in the $t$-channel:

$$V_{(2,1)|(1,1)} \times V_{(2,1)|(1,1)} \to V_{-2b\omega_1}(= V_{31|11}) \oplus \quad V_{-b\omega_1 - bh_2}(= V_{(1,2)|(1,1)}) . \tag{188}$$

Reminding (see 4.5) that at this value of the central charge we can identify the field $\omega_1$ and $\omega_2$ to $V_{(2,1)(1,1)}$ and $V_{(2,1)(1,1)}$, the final result is shown in (120). Notice that, when the central charge take the value $c = \frac{4}{5}$, not only one structure constants vanishes but also the $_3F_2$ hypergeomtric function entering the correlation simplifies in $_2F_1$ functions, that are solution of second order differential equation. This is explained by the fact that, for that particular value of the central charge, the monodromy group becomes reducible. This mechanism is analogous to the case of the energy operator $V_{(2,1)}^D$ in the Ising model. For general $c$, the four point function of $V_{(2,1)}^D$ satisfies a second-order differential equation. For $c = \frac{1}{2}$, the correlation simplifies to a solution of a first-order differential equation.

## C.3 Spin operators in the odd CFT?

Let us find out whether the fields $V^D, V^N$ (7) of the odd CFT can be linear combinations of the spin operators $\sigma_\alpha$ (95) of the Potts model, for $Q$ integer and in the scaling limit. We primarily characterize the fields $V^D, V^N$ by their two- and three-point functions (9)-(11). We moreover assume that the $\mathbb{Z}_2$ symmetry (8) corresponds to a $\mathbb{Z}_2$ subgroup of the $S_Q$ permutation symmetry of the Potts model.

In the scaling limit, we normalize the spin operators such that

$$\langle \sigma_\alpha \sigma_\alpha \rangle = Q - 1 , \tag{189}$$

which is equivalent to normalizing the two-point connectivity to one. Then the three-point connectivity is

$$\langle \sigma_\alpha \sigma_\alpha \sigma_\alpha \rangle = (Q-1)(Q-2)C , \tag{190}$$

where we assume that $C$ is given by Eq. (43). We look for expressions of $V^D, V^N$ as linear combinations of the type

$$V^D = \sum_{\alpha=1}^{Q} \lambda_\alpha^D \sigma_\alpha \quad , \quad V^N = \sum_{\alpha=1}^{Q} \lambda_\alpha^N \sigma_\alpha , \tag{191}$$

where the relation $\sum_\alpha \sigma_\alpha = 0$ allows us to assume

$$\sum_{\alpha=1}^{Q} \lambda_\alpha = 0 . \tag{192}$$



With this assumption, we can use the following technical results for two- and three-point functions of combinations of spin operators,

$$\left\langle \sum_\alpha \lambda_\alpha^{(1)} \sigma_\alpha \sum_\beta \lambda_\beta^{(2)} \sigma_\beta \right\rangle = Q \sum_\alpha \lambda_\alpha^{(1)} \lambda_\alpha^{(2)} , \tag{193}$$

$$\left\langle \sum_\alpha \lambda_\alpha^{(1)} \sigma_\alpha \sum_\beta \lambda_\beta^{(2)} \sigma_\beta \sum_\gamma \lambda_\gamma^{(3)} \sigma_\gamma \right\rangle = Q^2 C \sum_\alpha \lambda_\alpha^{(1)} \lambda_\alpha^{(2)} \lambda_\alpha^{(3)} . \tag{194}$$

For $Q = 2$, we cannot have a relation of the type (191), since the two independent fields $V^D, V^N$ cannot be expressed in terms of a unique spin variable. For $Q \geq 3$, let us assume that the $\mathbb{Z}_2$ symmetry corresponds to the permutation (12) of the spin variables. The behaviour under permutations determines $V^N \propto \sigma_1 - \sigma_2$, and implies $\lambda_1^D = \lambda_2^D$. If we assume $\lambda_{\alpha \geq 4}^D = 0$, we have the unique solution

$$V^D = \frac{1}{\sqrt{6Q}} (-\sigma_1 - \sigma_2 + 2\sigma_3) \quad , \quad V^N = \frac{1}{\sqrt{2Q}} (\sigma_1 - \sigma_2) . \tag{195}$$

This solution is such that

$$\left\langle V^D V^D V^D \right\rangle = \sqrt{\frac{Q}{6}} C , \tag{196}$$

which is the right result only if $Q = 3$. And in this case, the relations (195) is equivalent to the known relations (124). For $Q \geq 4$, there are other order two elements in $S_Q$. For $Q = 4$, the only element that is not equivalent to (12) is (12)(34). However, for $V^D$ to have the right behaviour under this permutation, we must have $V^D \propto \sigma_1 + \sigma_2 - \sigma_3 - \sigma_4$, which would imply $\left\langle V^D V^D V^D \right\rangle = 0$. We conclude that for $Q = 4$, there can be no linear expression for $V^D, V^N$ in terms of spin variables.

## Acknowledgements

We are grateful to Vladimir Dotsenko, Benoît Estienne, Yifei He, Yacine Ikhlef, Jesper Jacobsen, Nina Javerzat, Santiago Migliaccio, Slava Rychkov and Hubert Saleur, for helpful discussions. We thank the anonymous SciPost reviewers for their (publicly viewable) suggestions.

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
