# Peer review of "On four-point connectivities in the critical 2d Potts model"

_SciPost Physics, doi:SciPost Phys. 7, 044 (2019)_

## Round 1 · Referee Report · Anonymous (Referee 1) · 2019-7-26

Report

The authors extend their previous work and compare data for four-point cluster connectivities obtained in Monte-Carlo simulations with CFT predictions. These computations are done for the two-dimensional Potts model. Whereas very good numerical agreement is found with the predictions from the odd CFT, they provide arguments that this agreement is not exact for general values of Q.

This is an interesting paper that adds to the discussion of solving the Potts model for arbitrary values of Q. Based on the content I can recommend its publication. However, I do object to the format of the paper that is neither traditional, nor is it in agreement with the SciPost manuscript preparation guidelines. The authors made the choice to not provide all the necessary information to identify a reference, but instead mainly give the authors' names and the titles (and sometimes the arXiv number). References should allow the reader to quickly locate a relevant paper, but this is not possible without journal name, volume, first page (or paper identifier). In the first half of the paper figures and tables are not labeled nor are they accompanied by a descriptive caption. Figures in the second half of the paper (that present the numerical data) systematically do not have axes labels. The paper ends without a summary or discussion of the main results. The authors should adhere to accepted standards for a scientific publication before this manuscript is accepted.

Overall, the paper is written in good English, but there are still some typos (like 'phenomenons' and 'a four-point functions encodes" on page 2) that need to be corrected.

---

## Round 1 · Referee Report · Anonymous (Referee 2) · 2019-7-30

Report

The paper is devoted to four-point connectivities in the 2D Q-state Potts model at criticality. The problem of connectivities - probabilities for the distribution of N points among clusters - is fundamental for the theory of geometrical critical phenomena such as percolation (Q=1), but exact results have been missing for long time in the case of points in the bulk. New interest developed in the last years after that Delfino and Viti obtained the result for N=3 and studied the general relation with spin correlators, in particular for N=4. As part of these developments, the authors of the present paper mumerically implemented a conformal bootstrap for N=4 based on a conjecture for the spectrum of operators appearing in the operator product expansion [7]. The conjecture was supported by the numerical consistency of the bootstrap and by comparison with Monte Carlo simulations. More recently, however, Jacobsen and Saleur managed to show by a transfer matrix study that the spectrum conjectured in [7] is incomplete and proposed how to extend it. In the present paper the authors further compare Monte Carlo results for the N=4 Potts connectivities with their previous conjecture, which corresponds to a conformal field theory that they call the "odd CFT". They show that the latter provides results that agree with Monte Carlo data within the accuracy achievable in their simulations, but at the same time they confirm by analytic arguments that the four-point correlators of the odd CFT cannot be the exact answer for the Potts connectivities for Q generic, in particular for Ising (Q=2) and percolation. On the other hand, they argue that the answer is exact for Q=0,3,4, and show analytic coincidence with independent calculations for these values.

The paper is interesting and provides a further step in the study of an important and difficult problem. I have only few minor comments:

  1. The authors write at the beginning of section 3.3 that they will compare correlators of the odd CFT to Potts connectivities, and that they will also give an exact linear relation between them. I understand that they refer to Eq. (3.33), which however is not exact, as they stress after Figure 3.1. This point can be made more clear.

  2. The expansion (3.36) contains P_\sigma on both sides and can be written more clearly.

  3. The basic relation (1.2) calls for a reference (Dotsenko, Fateev, 1984).

  4. Eq. (2.16) extends results obtained for Liouville theory and its imaginary version, in particular by Zamolodchikov and Zamolodchikov. I suggest a reference to that work.

  5. Ref. [18] is at the origin of the developments of the last years on CFT of connectivities and should be quoted in the introduction.

---

## Round 2 · Referee Report · Anonymous · 2019-9-3

Report
The resubmission takes into account the comments contained in my report.
Anonymous on 2019-09-03 [id 587]
Editorial comment on item 1 in reply to anonymous report 1:
Indeed, titles are very helpful and Sci|Post is encouraging the inclusion of titles (see https://scipost.org/submissions/author_guidelines for the guidelines ). Beyond this, the authors are free to have their opinion, but they should still respect the standards of Sci|Post, in particular they should not create unnecessary work for a journal that is operating free of cost. Consequently, the authors will be expected to submit a version with all the required information (in particular the DOI for all references where one is available) prior to publication.

---

## Round 2 · Referee Report · Anonymous · 2019-9-15

Report
The authors have made the appropriate changes in the revised version of their manuscript. This is an interesting paper that contributes to understanding CFT as applied to these models. I recommend publication of the manuscript in its present form.

---

## Round 2 · Author Response

List of changes
Answer to the Anonymous Report 2:
1. In order to clarify this point, we have modified the beginning of Section 3.3 by using the word 'analytic' rather than 'exact' for the coefficients of the relation, and by adding the clause 'while stressing that the relation cannot be exact' at the end of the first paragraph.
2. At the end of Section 3.3, we have written dependences on Q explicitly.
3. For the basic equation (1.2), we have given [8] as a reference.
4. It is true that the structure constants of the odd CFT Eq. (2.16) are related to structure constants of Liouville theory. However, the relation is not simple or straightforward. Before and after (2.16), we have added references to the article [13], which specifically discusses the relation, rather than to works on Liouville theory.
5. We have cited the recommended Delfino-Viti article (now [6]) in Section 1.1, and added the sentence 'In particular, the three-point connectivity of the two-dimensional Potts model was found to be related to a three-point function in Liouville theory [6]'.
Answer to the Anonymous Report 1:
1. We have added clickable links to the specific versions of the cited articles that we have used in the two cases where they were missing. We disagree that journal references are needed to identify cited articles: using a search engine, titles are enough. Moreover, titles provide more useful information to the reader than journal references.
2. We have added captions to the figures and tables that did not have any. (Except a few small tables which are better treated as equations.)
3. We have added labels to the axes of the figures that did not have any.
4. There is no summary of the main results at the end of the paper because there is already a section 'introduction and summary' at the beginning.
5. We have corrected the typo 'a four-point functions encodes' on page 2. On the other hand, 'phenomenons' is a nonstandard but correct plural form of 'phenomenon'.
Further modifications:
We have added the reference [33] in Section A.1.

---

## Round 2 · List of Changes

Answer to the Anonymous Report 2:
1. In order to clarify this point, we have modified the beginning of Section 3.3 by using the word 'analytic' rather than 'exact' for the coefficients of the relation, and by adding the clause 'while stressing that the relation cannot be exact' at the end of the first paragraph.
2. At the end of Section 3.3, we have written dependences on Q explicitly.
3. For the basic equation (1.2), we have given [8] as a reference.
4. It is true that the structure constants of the odd CFT Eq. (2.16) are related to structure constants of Liouville theory. However, the relation is not simple or straightforward. Before and after (2.16), we have added references to the article [13], which specifically discusses the relation, rather than to works on Liouville theory.
5. We have cited the recommended Delfino-Viti article (now [6]) in Section 1.1, and added the sentence 'In particular, the three-point connectivity of the two-dimensional Potts model was found to be related to a three-point function in Liouville theory [6]'.
Answer to the Anonymous Report 1:
1. We have added clickable links to the specific versions of the cited articles that we have used in the two cases where they were missing. We disagree that journal references are needed to identify cited articles: using a search engine, titles are enough. Moreover, titles provide more useful information to the reader than journal references.
2. We have added captions to the figures and tables that did not have any. (Except a few small tables which are better treated as equations.)
3. We have added labels to the axes of the figures that did not have any.
4. There is no summary of the main results at the end of the paper because there is already a section 'introduction and summary' at the beginning.
5. We have corrected the typo 'a four-point functions encodes' on page 2. On the other hand, 'phenomenons' is a nonstandard but correct plural form of 'phenomenon'.
Further modifications:
We have added the reference [33] in Section A.1.

---

## Editorial Decision

published